# Resolving the controls of water vapour isotopes in the Atlantic sector

Jean-Louis Bonne [1], Melanie Behrens[1], Hanno Meyer [2], Sepp Kipfstuhl[1], Benjamin Rabe[1], Lutz Schönicke[2], Hans Christian Steen-Larsen[3,4] & Martin Werner [1]

Stable water isotopes are employed as hydrological tracers to quantify the diverse implications of atmospheric moisture for climate. They are widely used as proxies for studying past climate changes, e.g., in isotope records from ice cores and speleothems. Here, we present a new isotopic dataset of both near-surface vapour and ocean surface water from the North Pole to Antarctica, continuously measured from a research vessel throughout the Atlantic and Arctic Oceans during a period of two years. Our observations contribute to a better understanding and modelling of water isotopic composition. The observations reveal that the vapour deuterium excess within the atmospheric boundary layer is not modulated by wind speed, contrary to the commonly used theory, but controlled by relative humidity and sea surface temperature only. In sea ice covered regions, the sublimation of deposited snow on sea ice is a key process controlling the local water vapour isotopic composition.

[1] Alfred Wegener Institute, Helmholtz Centre for Polar and Marine Research, Bremerhaven 27570, Germany. [2] Alfred Wegener Institute, Helmholtz Centre for Polar and Marine Research, Potsdam 14473, Germany. [3] Geophysical Institute, University of Bergen and Bjerknes Centre for Climate Research, Bergen 5020, Norway. [4] Center for Ice and Climate, Niels Bohr Institute, University of Copenhagen, Copenhagen 2100, Denmark. Correspondence and requests for materials should be addressed to J.-L.B. (email: jean-louis.bonne@awi.de)

Stable water isotopologues $H_2^{18}O$ and $^1H^2H^{16}O$ undergo isotopic fractionation during phase transitions of water. Therefore, they can be used as integrated tracers of hydrological processes in the atmosphere. Their relative abundances compared with $H_2^{16}O$, expressed as $\delta^{18}O$ and $\delta^2H$ (see the Methods section), have been measured and used for many applications in climate-related studies, e.g., as proxies for past temperature[1] and precipitation[2,3] changes, variations of atmospheric moisture source conditions and transport pathways[4,5].

During phase changes, equilibrium and kinetic fractionation processes differently affect $\delta^{18}O$ and $\delta^2H$. The deuterium excess[6], hereafter d-excess, has been defined to quantify the kinetic effects (see the Methods section), such as those occurring during oceanic evaporation[6] or snow formation from supersaturated vapour at low atmospheric temperatures[7]. Merlivat and Jouzel[8], hereafter referred to as MJ79, developed a first theoretical model of isotope fractionation processes during evaporation from the ocean surface, which is still widely used. Applying their theoretical concept to the Earth's global water cycle, MJ79 introduced the so-called "closure assumption", assuming an equality of the isotopic composition of the net evaporated flux and the initial moist air above the ocean surface. According to this model, the strength of the d-excess signal in vapour is related to the relative humidity of the near-surface air with respect to the saturation vapour pressure at the ocean surface ($RH_{sea}$), as well as to the sea surface temperature (SST). The theoretical considerations by MJ79 led to different interpretations of past d-excess variations recorded in polar ice cores. They have been used as proxies of changes of the moisture source relative humidity[9] or SST[10–12]. The latter interpretation requires the assumption of negligible relative humidity variations during past climate changes, which has been recently challenged[13]. Regionally limited water vapour isotopic observations document a primary influence of the relative humidity on d-excess variability, while the influence of SST remains difficult to assess in this context[14–16]. Based on the evaporation theory and observations at the microphysical scale at the atmosphere–ocean interface, the model by MJ79 also considers an impact of wind speed on kinetic fractionation processes during evaporation and subsequently on the d-excess in the atmospheric vapour[8,17–19]. The importance of this wind-speed effect could not be validated so far by vapour d-excess observations performed only in coastal stations like Bermuda or Iceland[14,15], and is therefore still under debate.

In polar regions, variations in sea ice extent are supposed to affect regional precipitation amounts[20] and to be reflected in the water isotopic composition[21–23]. Current understanding of the impact of sea ice on the vapour isotopic composition is, however, still limited by the number of observations available in sea ice covered areas[24,25].

Here, we present a unique new dataset of ship-based in situ isotopic measurements of vapour and ocean surface water, conducted with an identical instrumental setup over 2 years for a large range of oceanic surface conditions at the basin scale of the Atlantic and Arctic Oceans, contrary to previous measurements more confined in area and time[25–27]. Our measurements, together with theoretical calculations and atmospheric simulations, allow for the first time the assessment of the variability of the water isotopic signal on the first order (the $\delta$ values of isotopic abundances for different species) and on the second order (the d-excess signal) under various climate conditions. For the process of oceanic evaporation, our dataset is consistent with the role of meteorological $RH_{sea}$ and SST, but rules out the theoretically assumed influence of wind speed on the d-excess of the initial vapour. Furthermore, the sublimation of snow deposited on top of sea ice is identified as a crucial process determining the near-surface vapour isotopic composition in sea ice-covered areas.

## Results

**Spatial and temporal variations.** Our observations, recorded onboard of the research vessel Polarstern, cover the period 29 June 2015 to 30 June 2017 and extend over a large range of latitudes in the Atlantic sector (i.e., Atlantic Ocean and the Atlantic regions of the Arctic and Southern Oceans), from the North Pole in the Arctic to the Weddell Sea in coastal Antarctica (see the Methods section for details).

All atmospheric measurements and simulated values are presented at a 6-h temporal resolution (see Fig. 1). The highest air temperature ($+28.6\,°C$), humidity ($19.3\,g\,kg^{-1}$) and $\delta^{18}O$ ($-8.4‰$) values are reported in the Inter Tropical Convergence Zone (ITCZ) in November 2015, April 2016, December 2016 and April 2017. Over open-sea regions, d-excess values are generally contained between $-10$ and $+10$ ‰, apart from rare short events up to $+15$ ‰, while in the ITCZ region, only positive values of d-excess are observed. Temperature, humidity and $\delta^{18}O$ progressively decrease from the ITCZ towards the mid and high latitudes of both hemispheres. In sea ice-covered polar regions, low $\delta^{18}O$ and high d-excess values are observed: in areas of compact sea ice coverage, minima in air temperature, specific humidity and $\delta^{18}O$ are reached ($-18.7\,°C$, $0.7\,g\,kg^{-1}$ and $-40.3‰$, respectively), together with a maximum in d-excess ($+22.3‰$). Similar extreme isotopic values are reported in August 2016 for a partial sea ice coverage, while the vessel was located in the vicinity of the Greenland ice sheet, close to the outlet of the Nioghalvfjerdsbrae glacier (latitude 79° N).

The atmospheric measurements have been accompanied by isotopic analyses of surface oceanic water samples (see Fig. 2), which depict strong latitudinal variations of the $\delta^{18}O$ signal in surface seawater. The distribution of $\delta^{18}O$ values is coherent to the GISS compilation[28]. The average $\delta^{18}O$ value for all our oceanic samples is $-0.7‰$. In the mid-latitudes, values between $-0.7$ and $0.9‰$ have been measured. The highest values are measured in the tropical bands, where evaporation dominates precipitation[29], with a maximum value of $1.1‰$ reached in the south tropical Atlantic, east of the Brazilian coast. In some parts of the Arctic region (western part of the Fram Strait, up to the North Pole), strongly negative values (down to $-5.4‰$) are measured, which could be due to the influence of the isotopically depleted waters originating from the large Siberian rivers, transported southwards along the eastern Greenland coast by the transpolar current. On the eastern part of the Fram Strait, as well as the Barents and Norwegian Seas, the $\delta^{18}O$ values stay similar to mid-latitude values, even in sea ice-covered areas (see Supplementary Fig. 3b). In the Antarctic sector, only slightly negative $\delta^{18}O$ values are observed (between $-1.6$ and $-0.4‰$), without any very isotopically depleted water masses contrary to the Arctic Ocean. The d-excess values of surface oceanic samples do not present clear latitudinal variations (see Fig. 2). The average d-excess value for all samples is of $2.2‰$. Only a few spots show slightly negative values (with a minimum of $-1.6‰$), while most of the samples present positive d-excess values (with a maximum of $8.1‰$).

Outputs from an atmospheric general circulation model with an explicit diagnostic of stable water isotopes (a so-called isoGCM) nudged to meteorology (see the Methods section) are compared with the observations. The atmospheric isotopic measurements are very well reproduced by one of the simulations (ECHAM$_{final}$) for the complete observational period (see Fig. 1). The data-model agreement is excellent for temperature (correlation coefficient $R^2 = 0.96$, Pearson correlation $p$-value $< 0.01$, for $N = 2389$ data points), specific humidity ($R^2 = 0.97$, $p < 0.01$, $N = 2513$), $\delta^{18}O$ ($R^2 = 0.82$, $p < 0.01$, $N = 2466$) and $\delta^2H$ ($R^2 = 0.85$, $p < 0.01$, $N = 2466$) and good for the second-order d-excess signal ($R^2 = 0.46$, $p < 0.01$, $N = 2466$).

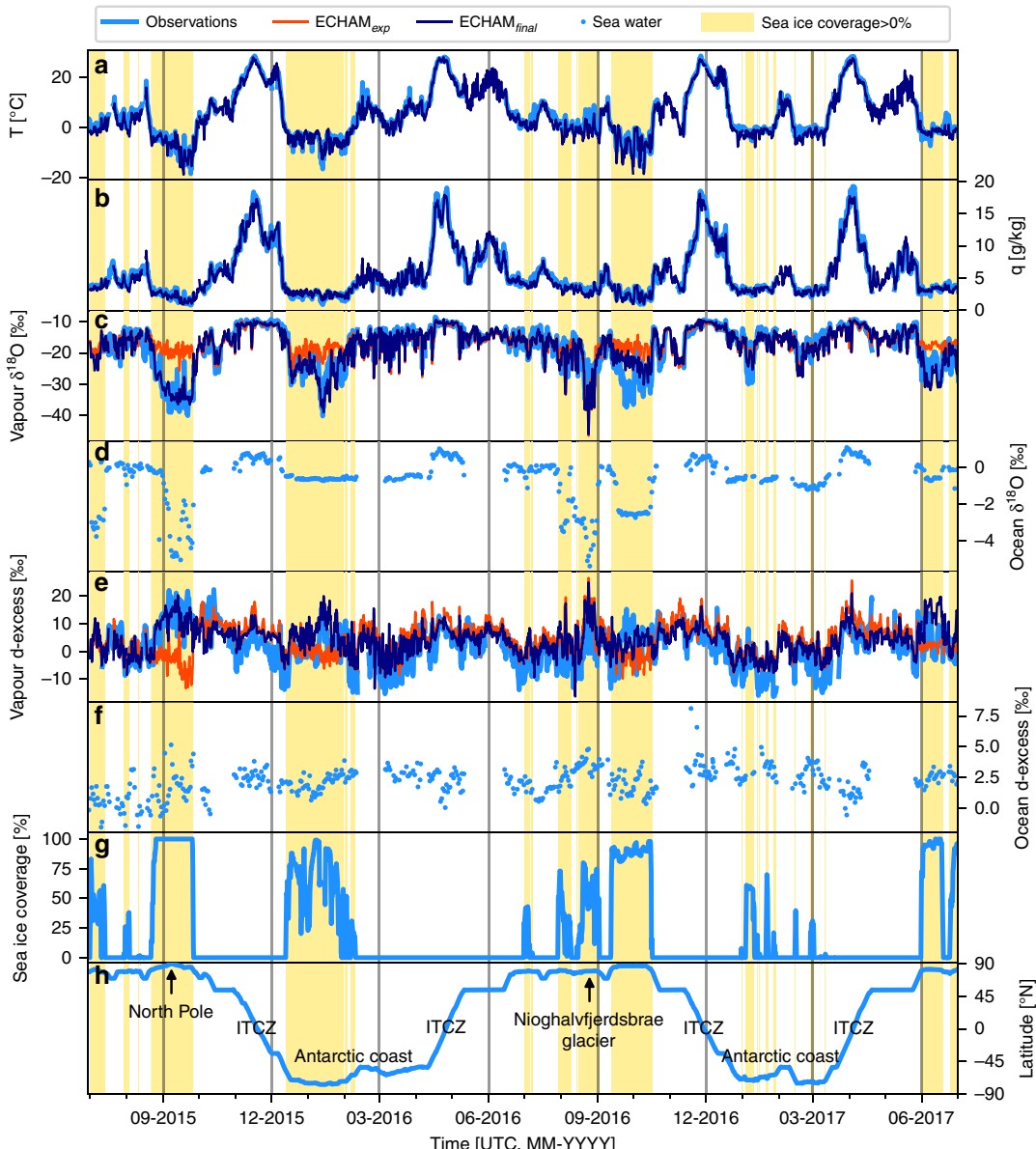

**Fig. 1** Observed and modelled water vapour and its isotopic composition in the Atlantic sector. Isotope measurements on-board of the research vessel Polarstern at 6-h resolution during the period 29 June 2015 to 30 June 2017. **a** Air temperature $T_{air}$ (°C); **b** specific humidity q (g kg$^{-1}$); **c** vapour $\delta^{18}$O (‰), **d** surface ocean water $\delta^{18}$O (‰); **e** vapour d-excess (‰); **f** surface ocean water d-excess (‰); **g** sea ice coverage (%); **h** latitude (° N). Pale yellow shades indicate periods with local sea ice coverage above 0%. For the air temperature, specific humidity, vapour $\delta^{18}$O and vapour d-excess panels (**a**, **b**, **c** and **e**), results from the different isoGCM simulations (thin lines) are shown for comparison, with the dark blue lines for the ECHAM$_{final}$ run and the orange lines for the ECHAM$_{exp}$ run. Note that both simulations give the same results for the temperature and specific humidity

**Deuterium excess controls during oceanic evaporation**. First analyses focus on the data obtained over open-ocean regions without land or sea ice upwind, within the observation period 29 June 2015 to 1 July 2017 (see the Methods section for details on the selection criteria). The corresponding dataset ($N =$ 1070 simultaneous vapour isotopic and meteorological observations) is distributed along the ice-free Atlantic region, from 81° N, near Svalbard, to 74° S in the Amundsen Sea (see Supplementary Fig. 3). Thus, the isotopic dataset and the related climate variables, e.g., SST and RH, are subject to both spatial and temporal (synoptic, seasonal and interannual) variations. Theoretical calculations derived from the MJ79 model and outputs from the isoGCM are used here to evaluate the evaporation processes influencing our observations.

A large range of meteorological conditions is covered by the observations from the open ocean, with very dry (+ 41%) to supersaturated (+ 125%) RH$_{sea}$ values and SST from −1.8 °C to + 29.1 °C. The correlation between both parameters is very low ($R^2 = 0.10$, $p < 0.01$). In our observations, the d-excess values in near-surface vapour are anti-correlated with RH$_{sea}$ and correlated with SST (with $R^2 = 0.62$ for RH$_{sea}$ and $R^2 = 0.50$ for SST, $p <$ 0.01). A multivariable linear regression of d-excess against both parameters indicates that constant d-excess values are distributed along oblique lines in an RH$_{sea}$/SST diagram (see Fig. 3). We obtain the empirically estimated function (with $R^2 = 0.76$ and a root-mean-square error of 3.4‰)

$$\text{d-excess} = -0.33 \cdot \text{RH}_{sea} + 0.27 \cdot \text{SST} + 25.01 \qquad (1)$$

**a**

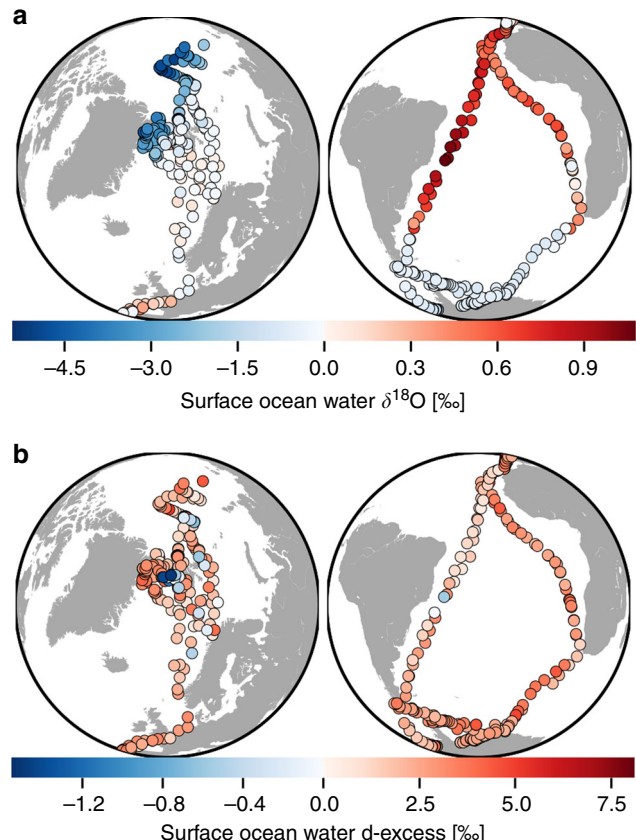

Surface ocean water $\delta^{18}O$ [‰]

−4.5 −3.0 −1.5 0.0 0.3 0.6 0.9

**b**

Surface ocean water d-excess [‰]

−1.2 −0.8 −0.4 0.0 2.5 5.0 7.5

**Fig. 2** Isotopic composition of surface ocean water during the period 29 June 2015 to 30 June 2017. The maps show the different measurement locations of surface ocean water from daily taken samples, with colours indicating the measured $\delta^{18}O$ and d-excess values in ‰, for panels **a** and **b**, respectively

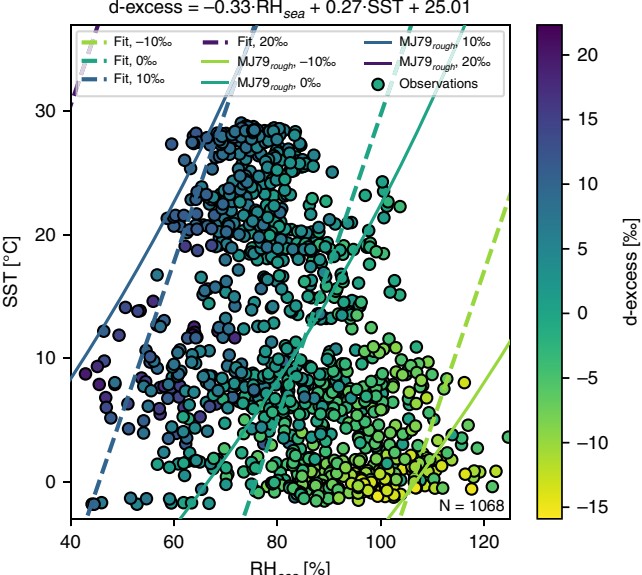

d-excess = −0.33·$RH_{sea}$ + 0.27·SST + 25.01

**Fig. 3** Observed influence of relative humidity ($RH_{sea}$) and sea surface temperature (SST) on deuterium excess (d-excess) in near-surface vapour compared with MJ79 theory. The scatters represent the observations in an SST/$RH_{sea}$ diagram, with colours indicating the d-excess values, for measurements at 6-h resolution during the period 29 June 2015 to 30 June 2017, considering only data without any potential influence of land- or sea ice-based processes (see the Methods section for details on data filtering). The dashed lines correspond to constant d-excess values (of −10‰, 0‰, 10‰ and 20‰ from green to purple lines) calculated from the empirical estimate of d-excess as a function of SST and $RH_{sea}$ (Eq. (1); see text for details). The solid lines correspond to constant d-excess values (of −10‰, 0‰, 10‰ and 20‰ from green to purple lines) calculated with the MJ79 model, for kinetic fractionation coefficients of a rough regime and the VSMOW-SLAP reference as the isotopic composition of the source

with d-excess in ‰, $RH_{sea}$ in %, and SST in °C. We note that this empirically estimated function includes both spatial and seasonal temporal changes in the evaporation conditions. While it covers a large range of meteorological conditions in the Atlantic sector, caution should be taken to apply this empirical function to isotopic data sampled in very different spatial or temporal domains.

To investigate the influence of the wind speed on the d-excess signal, we first focus on the distribution of observed d-excess values against wind speed (Supplementary Fig. 4). To filter out the primary control of $RH_{sea}$ and SST on the d-excess signal, we sort our observational dataset into several categories, where both $RH_{sea}$ and SST vary within a small range only. In each of these categories, no relationship can be observed between the wind speed and the d-excess values. Under the assumption that the measured d-excess values are caused by kinetic fractionation occurring during local oceanic evaporation, our results indicate that these fractionation processes are independent of the concurrent wind speed.

Next, we compare our open-ocean water isotopic measurements with calculations of the atmospheric boundary layer water vapour isotopic composition, based on the MJ79 evaporation model. In this model, an influence of wind speed on the kinetic fractionation during evaporation is considered, as wind will affect the surface roughness by generating waves, which in turn might alter the evaporation flux. Based on laboratory experiments[19], the MJ79 model assumes a smooth and a rough wind regime (below and above 7 m s$^{-1}$ surface wind speed, respectively) with distinct kinetic fractionation coefficients for both regimes. Three different

parameterisations of the kinetic fractionation are applied in our calculations. In the first parameterisation, a discontinuity is assumed in the kinetic fractionation coefficients at the wind-speed threshold of 7 m s$^{-1}$, as suggested by MJ79. The two other parameterisations use constant kinetic fractionation coefficients, identical to those applied either below or above this wind-speed threshold (see details in the Methods section). Our calculations imply a local closure assumption by considering only the local variations of $RH_{sea}$, temperature, wind speeds and oceanic surface water isotopic composition, neglecting any potential mixing of local vapour with advected air masses or convection.

The $\delta^{18}O$ and $\delta^2H$ values of all calculations are comparable, but do not match the observations (see Supplementary Fig. 5). The calculations always underestimate the short-term isotopic variations (related to synoptic variability) and overestimate the average isotopic levels compared with the observations. This overestimation can be explained by the applied local closure assumption, as the mixing of locally evaporated moisture with advected humidity is neglected in this model approach. For the MJ79 model, the closure assumption is in general not valid at the local scale, but at the global scale only[29]. The model may only yield the correct locally observed boundary layer $\delta^{18}O$ and $\delta^2H$ values if the atmospheric boundary layer was completely saturated with locally evaporated moisture. For our dataset, only the most enriched isotopic values observed in the low latitudes are matched by the MJ79 model estimates. At higher latitudes, the model values strongly overestimate the observed mean isotopic level. This latitudinal contrast might be due to the differences in

the proportions of moisture of local or advected origin, contributing to the local boundary layer humidity between the low and high latitudinal regions.

In contrast to $\delta^{18}O$ and $\delta^2H$, the atmospheric boundary layer variations in the d-excess signal are primarily controlled by kinetic fractionation processes occurring during evaporation. The wind-speed-related parameterisation of kinetic fractionation in the MJ79 model strongly affects the calculated d-excess values (see Fig. 4, Supplementary Fig. 6 and 7). Different kinetic fractionation coefficients used in the M79 calculations below or above the $7\,m\,s^{-1}$ threshold lead to different slopes in the distributions of d-excess versus $RH_{sea}$ for the two wind regimes (see Fig. 4). However, in our measurements, the distributions of d-excess against $RH_{sea}$ are nearly identical for both wind regimes (with 63% of wind-speed conditions above 7 $m\,s^{-1}$, see Fig. 5) and thus differ from the expected values from the MJ79 theory. For the three different parameterisations of the kinetic fractionation coefficients for the MJ79 calculation, using wind speed-dependent kinetic fractionation coefficients leads to the lowest agreement between observed and calculated d-excess values (slope of calculated versus measured values $m = 0.75$, $R^2 = 0.62$, $p < 0.01$; see Supplementary Fig. 7). With a parameterisation of the kinetic fractionation coefficients using the constant values of the rough wind regime, most observed d-excess values at open sea are correctly reproduced ($m = 0.65$, $R^2 = 0.67$, $p < 0.01$; see Supplementary Fig. 7). The calculation based on the kinetic fractionation coefficients of the smooth wind regime leads to an $RH_{sea}$/d-excess distribution (Fig. 4) with a similar slope as the observations, but is biased towards higher d-excess values (slope of calculated versus measured values $m = 0.94$, $R^2 = 0.71$, but with a $+ 4.9$‰ offset, $p < 0.01$, see Supplementary Fig. 7). We note that none of the parameterisations are able to reproduce the lowest measured d-excess values, corresponding to the highest $RH_{sea}$ values (see Fig. 4). Despite the overestimation of the first-order isotopic signals $\delta^{18}O$ and $\delta^2H$ in the local closure assumption, our observed d-excess variability can thus be reproduced by the MJ79 model approach, even on a local scale. The observations are better reproduced if constant kinetic fractionation coefficients are applied, and the best match between our data and the MJ79 model is achieved for the constant kinetic fractionation coefficients of a rough wind regime.

In the observations, the d-excess/$RH_{sea}$ distribution is characterised by a slope of $-0.39$‰ %$^{-1}$ ($R^2 = 0.64$, $p < 0.01$; see Fig. 5). In the MJ79-based calculations, this distribution is slightly different from the observations for any parameterisations of the kinetic fractionation, but the deviations are smaller when using the coefficients of a rough wind regime as compared with using the ones of a smooth wind regime. For the coefficients of a rough regime, the slope is of $-0.32$‰ %$^{-1}$ ($R^2 = 0.69$, $p < 0.01$; see Fig. 4), whereas it reaches $-0.5$‰ %$^{-1}$ ($R^2 = 0.84$, $p < 0.01$; see Fig. 4) for the coefficients of a smooth regime.

The impact of wind speed on the d-excess values of near-surface vapour is further evaluated through a sensitivity study using an isoGCM (see the Methods section for details). The isoGCM does not require any closure assumption, as it takes the mixing of locally evaporated vapour with advected moisture explicitly into account. Thus, it should in principle fit better to the observations than the MJ79 model calculations. For vapour over an open ocean, assuming two distinct evaporative regimes, with kinetic fractionation coefficients depending on the wind speed and a critical wind threshold of $7\,m\,s^{-1}$, gives rise to overestimated d-excess values compared with the observations (see Supplementary Figs. 6 and 8). This bias of ~5‰ disappears for the highest d-excess values when constant kinetic fractionation coefficients equivalent to a rough wind regime (see Supplementary Figs. 6 and 8) are applied. In both cases, the

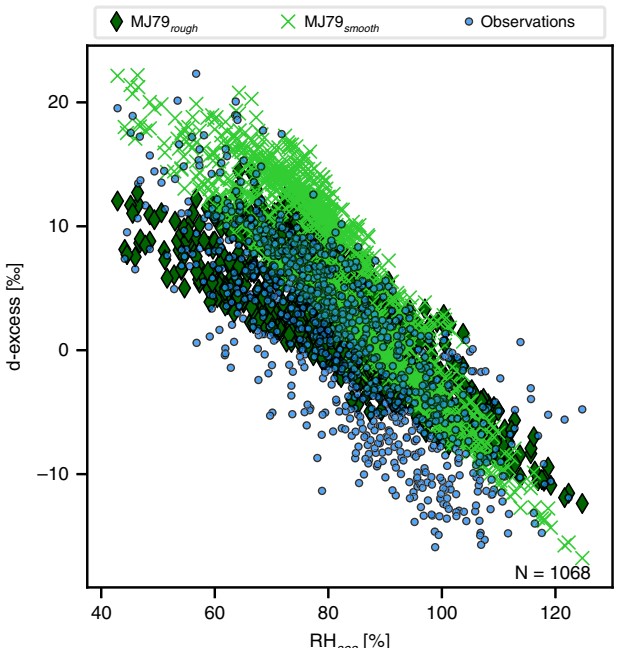

**Fig. 4** Sensitivity of the deuterium excess to $RH_{sea}$ values in observed vapour and in MJ79 model calculations with respect to kinetic fractionation parameterisation. Observations (light blue dots) are compared with results of the MJ79 model calculations using either the kinetic fractionation coefficients of a smooth wind regime (MJ79$_{smooth}$ calculation, light green crosses) or of a rough wind regime (MJ79$_{rough}$ calculation, dark green diamonds). Data for the period 29 June 2015 to 30 June 2017 over an open ocean only, at 6-h temporal resolution are used

lowest d-excess values are however overestimated and almost unaffected by the change of parameterisation.

The analysis of measured in situ d-excess values, the different calculations based on the MJ79 theoretical model and the simulations with a complex isoGCM all indicate that the variations of the atmospheric boundary layer d-excess values over the ocean surface are not modulated by wind speed, contrary to the suggestions made by MJ79. The d-excess values can be best explained by assuming constant kinetic fractionation coefficients in fractionation calculations, with the values that we originally used for a rough wind regime only.

Our results are based on observations within the boundary layer, ~30 m above the skin layer, at which the evaporation takes place. On the opposite, the wind speed-dependent kinetic fractionation parameterisation of MJ79 is based on wind tunnel experiments performed for a limited range of wind speeds and wave types[19]. From our analyses, we cannot make any conclusive statement about the validity of the model, as we did not performed comparable (laboratory) experiments directly above the water surface. However, our results clearly indicate that the MJ79 model should not be applied in its original form for the calculation of isotopic changes in atmospheric vapour well above the ocean, e.g., as done in current isoGCMs. The wind and wave-type range investigated for the MJ79 model approach might not necessarily represent the diversity of surface oceanic conditions observed at sea. For example, a rough ocean surface with high waves might also be caused by swell, and does not have to be directly linked to high wind speeds occurring at the same time. Based on our new dataset, we rather suggest to modify the MJ79 model and use constant kinetic fractionation coefficients instead of wind-speed-dependent values. This conclusion is supported by the recently reported lack of wind-speed influence on the water

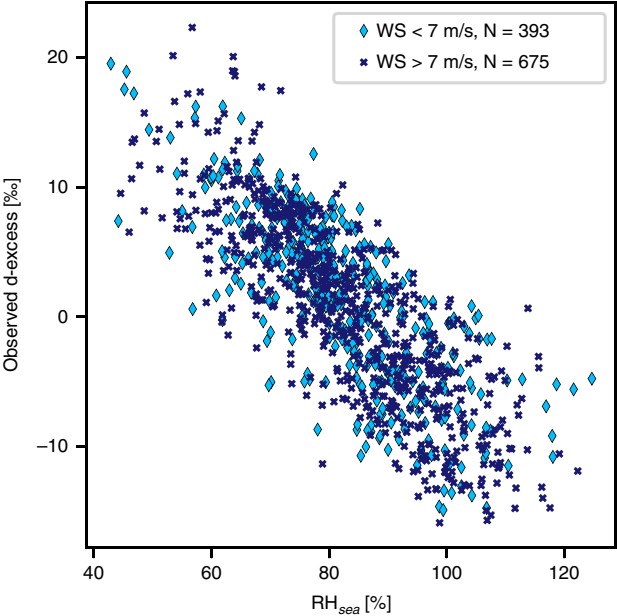

**Fig. 5** Sensitivity of the deuterium excess to $RH_{sea}$ values in observed vapour with respect to different wind-speed conditions. Values of d-excess observed at wind speeds below 7 m s$^{-1}$ (light blue diamond markers) are compared with values observed at wind speeds above 7 m s$^{-1}$ (dark blue crosses). Data for the period 29 June 2015 to 30 June 2017 over an open ocean only, at 6-h temporal resolution are used

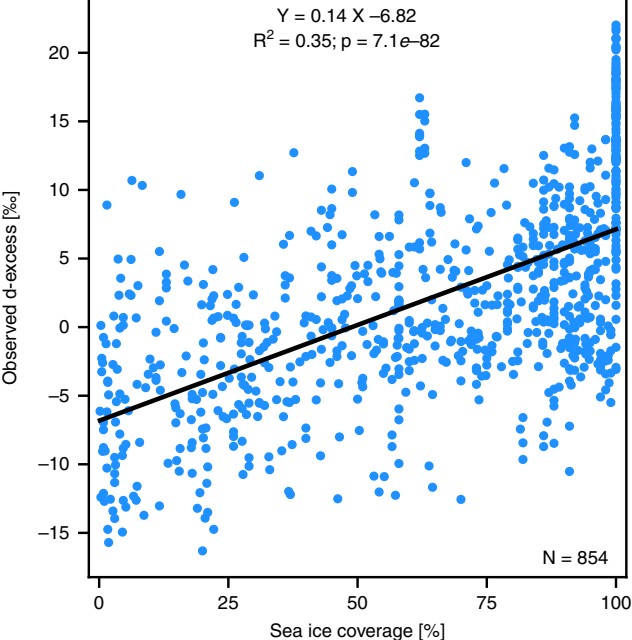

**Fig. 6** Sensitivity of the deuterium excess in vapour with respect to different sea ice coverage conditions. Relation between d-excess measurements and sea ice coverage around Polarstern for the period 29 June 2015 to 30 June 2017 (blue dots). Only d-excess measurements for sea ice coverages higher than 0% are considered. A linear regression is displayed in black. All plots are based on data with a 6-h temporal resolution

vapour d-excess signal measured at coastal stations in Bermuda and the south coast of Iceland[14,15].

### Influence of sea ice on water vapour isotopic composition.

Next, we focus on a subset of data gathered at high latitudes, where sea ice has been surrounding Polarstern (for local sea ice fractions higher than 0%, see the Methods section for details). The corresponding dataset contains measurements from both the Arctic and Antarctic regions ($N = 854$ simultaneous vapour isotopic and meteorological observations).

A recent study[24] postulated an anti-correlation between vapour d-excess and local sea ice fraction, based on near-ocean-surface vapour isotopic measurements, conducted over the course of approximately 3 summer days in the western Arctic. This anti-correlation was linked to the meteorological conditions at the sea ice margin. Our measurements cover a substantially longer time period and a larger spatial scale within both the Arctic and Antarctic sectors. They do not confirm such anti-correlation but rather indicate a positive correlation, with a d-excess increase of ~14‰ from open-ocean conditions to a complete sea ice coverage (see Fig. 6, Supplementary Fig. 9). However, the correlation between vapour d-excess and sea ice fraction is weak ($R^2 = 0.35$, $p < 0.01$) and does not improve when separating Arctic from Antarctic data for the analysis. A decrease of $RH_{sea}$ is observed with increasing sea ice coverage, related to an air temperature decrease, while the SST values cannot be lower than about $-1.8\,°$C (in complete sea ice-covered regions, the $RH_{sea}$ values are on average $-20\%$ lower compared with open-ocean conditions, but with a low correlation: $R^2 = 0.1$, $p < 0.01$; see Supplementary Fig. 9). In agreement with the kinetic fractionation theory of MJ79, this decrease of $RH_{sea}$ may partly contribute to this d-excess increase during oceanic evaporation, but applying the relationship observed over an open ocean, such $RH_{sea}$ variations can only explain half of the observed d-excess signal. The d-excess increase over sea ice-covered areas is also accompanied by a

depletion in $\delta^{18}$O and $\delta^2$H (on average of $-12‰$ in $\delta^{18}$O in complete sea ice coverage compared with open-ocean conditions, $R^2 = 0.34$, $p < 0.01$).

To identify the potential cause of this effect, we compare our measurements to two isoGCM simulations ($N = 840$ comparison points with observations; see Fig. 1 and Supplementary Fig. 10). In the first simulation, we assume that the isotopic composition of a bare sea ice surface is equal to the isotopic composition of the ocean water just beneath the sea ice, which is the usual procedure in such isoGCM simulations. For the second simulation, we assume that the isotopic composition of the sea ice surface is a function of the isotopic composition of a snow layer deposited on this surface (see the Methods section for details). Sublimation from the sea ice surface to the lowest atmospheric model layer is allowed in both cases, without considering any isotopic fractionation. In the first simulation, the modelled variations of $\delta^{18}$O and d-excess are small and do not agree with the measurements ($R^2 = 0.14$, $p < 0.01$ for $\delta^{18}$O; $R^2 = 0.00$, $p > 10^{-1}$ for d-excess, respectively; see Supplementary Fig. 8). In the second simulation, the measured low $\delta^{18}$O and high d-excess values of vapour over sea ice-covered areas are better simulated (see Fig. 1 and Supplementary Figs. 8 and 10). Spatial and temporal variations of both parameters are reproduced ($R^2 = 0.6$ for $\delta^{18}$O, $R^2 = 0.35$ for d-excess, $p < 0.01$, see Supplementary Fig. 8) for measurements in both hemispheres. We conclude that the snow accumulated on top of sea ice, which has depleted $\delta^{18}$O and $\delta^2$H, and high d-excess values compared with seawater, is a potential additional key source determining the atmospheric boundary layer vapour isotopic composition in sea ice-covered regions. We note that the applied parameterisation of the fraction of sea ice covered by deposited snow (see the Methods section) is based on a subset of our observational data and thus does not represent a strict independent proof for the importance of snow sublimation as a source for the isotopic composition of the vapour. We rate it as a first-order approach to include snow on sea ice for

future isotope modelling studies. Further observational data are certainly necessary to validate and improve this parameterisation, e.g., to take the flushing of the snow by seawater in fragmented sea ice areas into account, as well as potential isotopic fractionation effects during the sublimation of the snow.

During August 2016, measurements on the research vessel have been performed in an area with only a partial sea ice coverage in the vicinity of the Greenland ice sheet, close to the outlet of the Nioghalvfjerdsbrae glacier (latitude 79° N). Very depleted isotopic values of near-surface vapour measured during this period ($\delta^{18}$O reaching a local minimum of −37.7‰ close to the values observed at NEEM on top of the Greenland ice sheet[30]) are matched by both isoGCM simulations, independently of the parameterisation of sublimation above sea ice. Advection of isotopically depleted vapour from the Greenland ice sheet towards the research vessel could create a signal overprinting the local vapour isotopic composition. The model simulates sublimation over Greenland with the same surface source for both parameterisations, contrary to the sublimation taking place on the sea ice, and would provide the same isotopic composition in both simulations. However, such influence of katabatic winds on our dataset is generally limited, both around Greenland and Antarctica. Within the sea ice-covered area, air masses originating from coastal regions, as filtered for the open ocean, represent ~8% of the dataset. Our results concerning the sea-ice influence on d-excess do not change when filtering data points potentially influenced by such continental air masses (not shown here).

## Discussion

Our results are based on direct isotopic measurements, on calculations applying the MJ79 model and on results from complex isoGCM simulations. Our measurements support the fundamental theory of kinetic fractionation by MJ79[8] concerning the influences of both relative humidity and temperature at the atmosphere–ocean interface on the atmospheric boundary layer d-excess of vapour over the oceanic surface. However, contrary to this theory, our data suggest that the kinetic fractionation is not modulated by wind speed. Considering constant fractionation coefficients with values for a rough wind regime yields best agreement between observed and modelled d-excess values. The general relationship we obtain for the distribution of d-excess as a function of relative humidity and SST is based on a compilation of observations from various climatic regions, ranging from the tropics to high latitudes. For the calculation of this relationship, we neglected the potential influence of advected moist air on our measured data. Thus, the relationship should be used with care for oceanic regions, where moisture advection might substantially contribute to the boundary layer water vapour content. For sea ice-covered regions, our results indicate that sublimation of snow on sea ice might be a key additional process, controlling the isotopic composition of the boundary layer water. This vapour can subsequently influence the isotopic signal of polar precipitations.

Hence, our results have, among others, the following implications for paleoclimate studies based on water isotope records, e.g., derived from ice cores and speleothems, as well as for present-day hydrological studies. Firstly, the variations of d-excess should be interpreted as a mixed proxy for both relative humidity and SST conditions at the moisture source, but not as a proxy for wind speed. In this regard, a 10% increase in $RH_{sea}$ would reduce the d-excess by ~3‰, while a 10 °C increase in SST would raise the d-excess by about 3‰. Secondly, at high latitudes, isotopic variations in near-surface vapour are strongly influenced by evaporated ocean water, but potentially also by a snow cover on the sea ice, which has an isotopically different source signal than ocean

water. Combined with the decrease of relative humidity towards sea ice-covered areas, this leads to an ~1.2‰ decrease in $\delta^{18}$O and 1.4‰ increase in d-excess for every 10% increase in sea ice coverage. This sea ice effect in $\delta^{18}$O, $\delta^2$H and d-excess may have an imprint on the subsequent water isotopic composition of precipitations. It may then contribute to explain, for instance, some abrupt variations of the d-excess signal recorded in Greenland ice cores at the end of the last glacial period[31] or to validate a hypothesis of past sea ice retreat at 128 ka around the West Antarctic Ice Sheet[32]. Water isotopic variations in ice cores may also be used as a proxy for regional sea ice extent in the Arctic and Antarctic sectors, in combination with other chemical proxies[33]. For this purpose, it is needed to carefully evaluate the moisture source locations of the sites where the ice cores are retrieved, as well as potential post-depositional processes affecting d-excess values in the firn layer. Another implication of our results concerns the parameterisation of future isoGCM simulations focusing on polar regions, which should explicitly consider the snow on top of sea ice, identified as a potential additional sublimation source affecting the isotopic signal. The first-order parameterisation deduced from our observational data and isoGCM experiments might be used for such modelling studies, but further independent observational data and simulation results are certainly required for improving this parameterisation.

## Methods

**Meteorological observations.** Routinely measured meteorological data from Polarstern are used in this study. The related sensors are located at different heights: wind speeds and wind directions are measured at 39 m above sea surface, relative humidity ($RH_{air}$) and temperature ($T_{air}$) at 29 m above sea surface and water temperature (SST) at 5 m below sea surface. Air pressure (P) is measured at an altitude of 19 m, but expressed at sea level. The calibrated and validated datasets are available at a 10-min averaged temporal resolution on PANGAEA Open Access library[34] and have been averaged at a 6-h temporal resolution in this study.

The relative humidity of the near-surface air with respect to the saturation vapour pressure at the ocean surface ($RH_{sea}$) is not directly measured but has been approximated[27,35] from the observed relative humidity $RH_{air}$ at 29 m, corrected by the ratio of specific humidity at saturation between the temperatures at this elevation and at the sea surface ($T_{air}$ and SST)

$$RH_{sea} = RH_{air} \cdot \frac{q_{sat}(T_{air})}{q_{sat}(SST)} \tag{2}$$

where $q_{sat}(T)$ is the specific humidity at saturation for a given temperature T and $q_{sat}(SST)$ is calculated for seawater at salinity 35 PSU[36]. For intercomparison with other sea surface water vapour isotopic measurement campaigns[27], this calculation is performed using the air temperature and relative humidity corrected from the altitude at 10 m.

**Sea surface temperatures.** For the complete measurement period, the skin SSTs (sea surface temperatures adjusted to compensate for a skin temperature bias above a wind speed of 6 ms⁻¹) are retrieved at the Polarstern locations from the Met Office Operational Sea Surface Temperature and Ice Analysis (OSTIA)[37] products at a 0.25° × 0.25° horizontal resolution. The original dataset has a 1-h temporal resolution and has been averaged at a 6-h temporal resolution. A comparison with the Polarstern SST measurements at 5-m depth for the period 29 June 2015 to 31 January 2017 gives a very good agreement between both datasets (SST$_{OSTIA}$=0.99·SST$_{Polarstern}$−0.01 [°C]; $R^2$=0.99; $p < 0.01$; N = 2099).

**Sea ice coverage.** Due to a lack of continuous and quantitative sea ice observations during the different Polarstern cruises, the sea ice coverage surrounding the research vessel has been derived from ERA-interim reanalyses[38] at 0.75° × 0.75° spatial and 6-h temporal resolution. The sea ice coverage at a specific Polarstern location is assumed to be equal to the value of the surrounding grid cell. This dataset has been compared and is coherent with values extracted in the same manner from daily sea ice coverage data from the AMSR2 instrument on-board the GCOM-W1 satellite at a 6.25-km resolution.

**δ-notation for isotopic composition.** Isotopic compositions of samples are reported using the δ-notion, where $R_{sample}$ and $R_{VSMOW}$ are the isotopic ratios ($H_2^{18}O/H_2^{16}O$ or $^1H^2H^{16}O/H_2^{16}O$ for $\delta^{18}$O and $\delta^2$H, respectively) of the sample

and of the Vienna Standard Mean Ocean Water (VSMOW2)[39], respectively:

$$\delta = 1000 \cdot \left( \frac{R_{sample}}{R_{VSMOW}} - 1 \right) \quad (3)$$

**Definition of deuterium excess**. The deuterium excess values are computed based on the commonly used definition[6]:

$$\text{d-excess} = \delta^2H - 8 \cdot \delta^{18}O \quad (4)$$

**Water vapour isotopic composition**. A Cavity Ring Down Spectroscopy (CRDS) analyser (model L2140-i, Picarro, Inc.) has been running continuously on-board of the research vessel Polarstern since the 29 June 2015, recording humidity mixing ratio, $\delta^{18}O$ and $\delta^2H$ values of water vapour at a temporal resolution of ~1 s. The ambient air inlet for this instrument is located at 29 m above the sea level, connected to the analyser through an ~25 -m-long tubing heated at 65 °C. The humidity mixing ratio is converted into specific humidity measured by the CRDS analyser ($q_{CRDS}$) and corrected by a linear function derived from the direct comparison with specific humidity values derived from the meteorological observations ($q_{meteo}$) on-board the Polarstern, during the complete measurement period from 1-h resolution datasets: $q_{meteo} = 0.75 \times q_{CRDS} - 0.17$ ($R^2 = 1.0$, $p < 0.01$, for $N = 17592$). $q_{meteo}$ is calculated based on $RH_{air}$, $T_{air}$ and P. The precision of specific humidity measurements is estimated at 0.1 g kg$^{-1}$ from the comparison of both datasets.

For instrument calibration of the isotopic values, a custom-made system is used, vaporising water isotopic standards injected in liquid form and mixed with dry air provided by high-pressure gas cylinders. Four different liquid isotopic standards are used, covering the range of the expected ambient air values ($\delta^{18}O$ values between −7.8‰ and −40.7‰). Recommendations for long-term calibration of CRDS water vapour isotopic analysers were followed[40,41]. Therefore, our system allows two types of calibration. Firstly, the concentration dependence[34,35] of the raw isotope measurements is corrected. Secondly, repeated corrections of the deviation of the measurements from the VSMOW-SLAP scale[42] are performed by the computation of calibration curves based on the measurements of the water standards, thereby allowing correction of the instrumental drift.

The humidity-concentration dependence of the isotope measurements is corrected based on the measured isotopic composition of each four water standards over a range of different humidity values. The results of the calibration measurements are presented in Supplementary Fig. 1. The temporal stability of this correction has been evaluated by successive measurements of this so-called humidity-response function at different times. No significant drift of this response was observed for any of the four standards, neither for successive measurements over a week (not presented separately in the graphics) nor for measurements separated by several months over the complete observational period (as shown by the different measurement sequences in the graphics). The humidity-response function is thus considered constant in time. It does however depend on the isotopic standard used. A humidity-response function is computed for each isotopic standard as the interpolation of the distribution of all experiments with a polynomial function of fourth order. The correction of the humidity-concentration dependence for a specific near-surface vapour measurement is determined by the linear interpolation of the two humidity-response functions from the closest surrounding isotopic standards at the isotopic value of the measurement.

Calibration curves are applied to the raw data to correct deviations from the VSMOW-SLAP scale. These calibration curves are calculated based on the repeated measurement of every liquid standard for 30 min every 25 h (a standard measurement sequence consists of the successive measurement of all four calibration standards). To avoid any memory effects, averaged values and standard deviations of the standard isotopic composition are computed over the last 15 min of each injection only. Several filtration and correction steps (summarised in Supplementary Table 1) are applied to these standard measurements before computing the calibration curve. All measurements are corrected for the humidity-concentration dependence. To account for the difference in the isotopic composition of the same liquid standard stored in different bottles and used on separate injection lines, we define an arbitrary reference standard among both bottles and correct the measured isotopic values from the difference between the known isotopic value of both bottles. We remove measurements with average $H_2O$ values ($\overline{H_2O}$) below 5000 ppm or higher than 28000 ppm and standard deviations of $H_2O$, $\delta^{18}O$ or $\delta^2H$ (noted $\sigma(H_2O)$, $\sigma(\delta^{18}O)$ and $\sigma(\delta^2H)$, respectively) higher than 2500 ppm, 1.5‰ and 5‰. We compute a first 14-day running average and eliminate all measurements that deviate from this running average by more than 1.5‰, 5‰ and 8‰ for $\delta^{18}O$, $\delta^2H$ and d-excess, respectively. The observed variabilities of these selected measurements of all liquid standards are shown in Supplementary Fig. 2.

The calibration curves are calculated every time a standard measurement sequence has been performed, based on a new 14-day running average of the previously selected liquid standard measurements. These values are compared with the theoretical values of the reference standards at the time of the standard measurement sequence. If values of at least 3 standards are available, a linear regression of the measurements against the theoretical values gives the calibration curve. Otherwise, as found in the literature for such type of analysers[14], we correct the calibration curve from the drift of the running average of the standards, which

have been correctly measured and use an interpolated value of the slope between the closest calculated calibration curve.

Based on the uncertainty of both corrections from the concentration dependence and deviations from the VSMOW-SLAP scale, the measurement accuracy is estimated at 0.16‰, 0.8‰ and 2.1‰ on $\delta^{18}O$, $\delta^2H$ and d-excess. The measurement precision on 1-h averages, estimated from the standard deviation of calibration standard measurements at a constant humidity level, is of 0.24‰, 0.7‰ and 2.7‰ on $\delta^{18}O$, $\delta^2H$ and d-excess for humidity levels above 5 g kg$^{-1}$. It deteriorates with lower humidity levels, reaching 0.5‰, 1.9‰ and 5.9‰ for $\delta^{18}O$, $\delta^2H$ and d-excess, for humidity levels of 1 g kg$^{-1}$. The dataset presented in this study has been averaged at a 6-h temporal resolution.

**Surface water isotopic composition**. Isotopic composition of the surface oceanic water has been measured from daily taken samples since 30 June 2015, filled in narrow-mouth low-density polyethylene 20- or 30-mL plastic bottles, sealed with Parafilm M and stored at +4 °C from the end of the expedition until the measurement. Measurements are done with IRMS and equilibration technique at the isotope laboratory of AWI Potsdam[43] (with an accuracy better than 0.1‰ and 0.8‰ for $\delta^{18}O$ and $\delta^2H$). For comparison with other parameters, an interpolation of this dataset has been used at a 6-h resolution.

**MJ79 model under the closure assumption**. For all the observations performed above the open ocean, we compute the corresponding theoretical water vapour isotopic composition in the atmospheric boundary layer over an open ocean based on the MJ79 model under the closure assumption. We assume it to be equal to the isotopic composition of the evaporation flux[8,35]. We thus express the boundary layer vapour isotopic ratio ($R_{BL}$) as a function of the surface seawater isotopic ratio ($R_{SW}$), taking both equilibrium and kinetic fractionation coefficients $\alpha_{eq}$ and $\alpha_k$ and $RH_{sea}$ into account:

$$R_{BL} = \frac{R_{SW}}{\alpha_{eq} \cdot (\alpha_k + RH_{sea}(1 - \alpha_k))} \quad (5)$$

We use skin temperature at the air–sea interface from the OSTIA dataset to determine $\alpha_{eq}$ values[44]. $R_{SW}$ is determined by the interpolated values of the isotopic composition measured in daily sampled surface oceanic water.

We use three different parameterisations for the kinetic fractionation coefficients dependency on wind speed. In the first simulation (named MJ79$_{ref}$), the kinetic fractionation coefficients ($\alpha_{k,^{18}O}$ and $\alpha_{k,^2H}$ for $H_2^{18}O$ and $^1H^2H^{16}O$) for a smooth or a rough wind regime are used for wind speed, respectively, below or above the threshold of 7 m s$^{-1}$. We respectively apply kinetic fractionation coefficients[35,45]: $\alpha_{k,^{18}O} = 1.0060$, $\alpha_{k,^2H} = 1.0053$ for a smooth wind regime; $\alpha_{k,^{18}O} = 1.0035$, $\alpha_{k,^2H} = 1.0031$ for a rough wind regime. We use the values of the measured wind speed on Polarstern (at 39 m above sea surface). In two additional simulations (respectively named MJ79$_{smooth}$ and MJ79$_{rough}$), the kinetic fractionation coefficients are set constant and independent of the wind speed, either to the value of a smooth or a rough wind regime.

**Atmosphere general circulation model with water isotopes**. In this study, isoGCM simulations are performed with the ECHAM5-wiso model[46] with a horizontal grid size of ~1.1×1.1° (T106 spectral resolution) and 31 vertical levels. The model is nudged to ERA-interim surface pressure, temperature, vorticity and divergence fields[47] to ensure that the simulated large-scale atmospheric flow is modelled in agreement with the ECMWF reanalysis data on all analysed timescales during the years 2015–2017. For each time step of 6 h, isoGCM simulation results of near-surface vapour amount and its isotopic composition are extracted from the model grid cell encompassing the position of Polarstern. In the vertical direction, this grid cell extends from the surface to ~60 m above the surface. Two different ECHAM5-wiso simulations are performed, all covering the period from January 2015 to July 2017 after a 12-month spin-up period.

In the first simulation (named ECHAM$_{exp}$), different kinetic fractionation coefficients during evaporation over open water are applied depending on wind speed: constant coefficients for a smooth wind regime, and wind speed-dependent coefficients for a rough wind regime are used for wind speeds below or above the threshold of 7 m s$^{-1}$, respectively[48]. Over sea ice-covered areas, bare ice is prescribed with an isotopic composition of ocean surface water, based on a global gridded data compilation of $\delta^{18}O$ in seawater[49].

In the second simulation (named ECHAM$_{final}$), constant fractionation coefficients for $\delta^{18}O$ and $\delta^2H$, suggested for a rough wind regime[35], are applied under all different meteorological conditions for evaporation processes. Over sea ice-covered areas, a 2 -cm-deep snow layer pad is assumed on top of any sea ice-covered grid-cell fraction, to account for accumulation and sublimation of snow on sea ice. The isotopic composition of the bare sea ice is assumed to be equal to the composition of surface ocean waters, neglecting a potential small fractionation process occurring during the formation of sea ice. The prescribed surface ocean $\delta^{18}O$ and $\delta^2H$ values are taken from a reference global gridded dataset compilation[49]. The isotopic composition of the snow layer is determined by the isotopic composition of the accumulated snowfall. During sublimation processes, no fractionation of the snow is assumed. This treatment of snow on sea ice as a single-layer bucket model is equivalent to treatment of snow on land surfaces in

ECHAM5-wiso. The deposited snow in the model is locally controlled, without taking any advection of sea ice or snow drift into account. In reality, the isotopic composition of the sea ice surface will not only be determined by the isotope signal of bare ice or snow on top of the sea ice, but also by further processes altering the sea ice surface. For example, sea spray or breaking waves might substantially alter the isotopic composition of a snow-covered sea ice surface, especially for regions with only a minor area fraction covered by sea ice. These effects will lead to a further mixing of the isotopic signal of the original fallen snow with the isotopic composition of the surrounding ocean surface waters. To account for such processes in ECHAM5-wiso, the isotopic composition of the sea ice surface is assumed as

$$\delta_{\text{sea ice surface}} = f^4 \cdot \delta_{\text{snow bucket}} + \left(1 - f^4\right) \cdot \delta_{\text{ocean}} \qquad (6)$$

with $\delta$ as $\delta^{18}O$ or $\delta^2H$, $\delta_{\text{sea ice surface}}$ the isotopic composition of the sea ice surface, $\delta_{\text{snow bucket}}$ the isotopic composition of the snow bucket on sea ice, $\delta_{\text{ocean}}$ the isotopic composition of the surrounding ocean surface water and f as the fraction of sea ice in each grid cell. This empirical formula is based on a comparison of measured and simulated $\delta^{18}O$ and d-excess values for the period from August to October 2015 and applied for the data-model comparison over the whole measurement period of this study (see Fig. 1) afterwards.

**Data filtering**. For analysing evaporation processes occurring over an open ocean, without any potential influence of land- or sea ice-based processes, we filtered all the data with sea ice or land situated upwind of each measurement. The upwind area is defined by a 40° angle centred around the wind origin and a maximum distance of vapour transport within 14 h previous to the measurement, which is determined by the measured Polarstern wind speed. Only if this area is free of both sea ice and land (0% sea ice index and no land area), we consider the corresponding measurements as influenced by surface processes over the open ocean. Vice versa, the isotope data corresponding to sea ice-covered areas are selected by including all measurements, where the ERA-interim sea ice coverage in the grid cell surrounding the research vessel Polarstern is higher than 0%. The locations of all filtered datasets are displayed in Supplementary Fig. 3a, b.

## Data availability
All presented instrumental and modelling data of this study are available on the PANGAEA database[50].

## Code availability
The code of the ECHAM model can be retrieved from the Max-Planck-Institut für Meteorologie and is subject to a license. The isotope enhanced version is available by personal contact to the authors.

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

## Acknowledgements

This study has been funded by the AWI Strategy Fund project ISOARC. The measurements on-board Polarstern research vessel were conducted during the PS93.1 (grant no. awi-ps9301), PS93.2, PS94, PS95.1, PS95.2, PS96, PS97, PS98, PS99.1, PS99.2, PS100, PS101, PS102, PS103, PS104, PS105, PS106.1 and PS106.2 expeditions. We deeply acknowledge the different persons who took part in the maintenance of the instrument during these campaigns: Sandra Tippenhauer, Mario Hoppmann and Hendrik Hampe, Ronny Engelmann, Stephanie Bohlmann, Stefanie Arndt, Leonard Rossmann, Lester Lembke-Jene, Vera Schlindwein, Ole Valk, Myriel Horn, Mooritz Haarig, Heike Kalesse, Hendrik Hampe, Elke Burkhart, Michael Flau, Boris Christian, Julia Goedecke, Anna Nikolopoulos, Torsten Linders and Céline Heuzé.

## Author contributions

All authors contributed to the design of this study. Instrument layout and Picarro installation on Polarstern was done by J.-L.B., M.B., H.M., S.K., L.S., H.C.S.-L. and M.W. Isotope measurements and instrument maintenance were performed by J.-L.B. and M.B. Ocean isotope sampling on-board of Polarstern was advised by B.R. and ocean isotope measurements were done by H.M. IsoGCM simulations were performed by M.W. The first paper draft was written by J.-L.B. and M.W., and all authors contributed to the discussion of the results and the final article.

## Additional information

**Competing interests:** The authors declare no competing interests.

