## [Peer Review File · Nature Communications]

Reviewer #1 (Remarks to the Author):

Increasingly stable water isotopes are being seen as a key component of analysing many aspects of the workings of the global circulation, and particularly the transport of water vapor. The focus here is on deuterium-excess and what it reveals about the local boundary layer over the ocean. The authors posit that this excess is controlled by the relative humidity and ocean temperature, and not by wind speed. They argue that the various influences (and their relative importance) on d-excess have in the past only been determined in atypical locations such as coastal Iceland and Bermuda.

The authors present new dataset of in situ isotopic measurements of ocean surface water and vapor which was obtained from an extensive Polarstern cruise covering the global Atlantic sector. Data on the cruise were collected every 6 hr over the period mid-2015 to mid-2017.

This manuscript is potentially a valuable contribution to this understanding. However, I outline below some aspect in which the submission should be revised.

Lines 79-81: Figure 1 presents a neat overview of the observations and the location (latitude) of Polarstern. The authors should make clear how the 'average values' were calculated. If this was done with the 6-hr data we would not get a true picture of the Atlantic average, as the ship spends most time in the polar regions, and relatively little time equatorward of 45 degrees (as deduced from bottom panel of Fig.). Please clarify. As a broader comment, I am not sure whether these average values have much meaning.

Also, in the Fig. it is hard to differentiate between the ECHAMexp and ECHAMfinal curves. Can you make this easier for the reader (e.g., increase the contrast, use dash lines etc.)

Line 61: Make reference here also to the analysis of Noone et al., 2004: Sea ice control of water isotope transport to Antarctica and implications for ice core interpretation. *J. Geophys. Res.*, 109, D07105, doi:10.1029/2003JD004228.

Line 91 (and throughout): It would be appropriate to refer to the 79 deg glacier as Nioghalvfjærdsbrae.

Lines 94-96: This statement is vague and misleading. Perhaps refer to one of the standard global climatologies (e.g., the GISS product) which show peak values of oceanic $\delta O18$ at about 20degrees latitude (just equatorward of the subtropical ridge in both hemispheres. The key factor here is P-E, not evaporation.

Lines 115-127: A problem I have with the discussion of the correlations of the 'baseline' values is that the data are varying in both space and time. If two variables are correlated in the space domain this, a priori, tells us nothing about how they might be correlated in time. A spatial relationship between two variables in general may say nothing about the temporal relationship. The classic Dansgaard 1964 study showed a strong annual SPATIAL relationship between T and $\delta^{18}O$, the SEASONAL cycle of parameters often yields associations opposite to those of the spatial analysis. For example, $\delta^{18}O$ in summer precipitation at Lhasa (Tibetan Plateau) is depleted by more than 6 ppt with respect to winter rainfall, despite the surface air temperature being 10°C higher in summer. (This is because gradual rain-out of moist, oceanic air masses moving inland, associated with the monsoon circulation, constitutes a powerful mechanism capable of producing large isotopic depletions in rainfall, often completely overshadowing the dependence of them on temperature.)

Please word this part of the text more carefully.

In a similar vein the regressions shown in Figure 3 reflect a mixture of time and space sampling, and the authors should make clearer the physical argument they are presenting. I am reminded of the study of Brown et al., 2004: The dependence of the $\delta^{18}O$ -temperature relationship on continent locations. *Geophys. Res. Lett.*, 31, L09208 who conducted experiments with an isotope-enabled GCM. He found that by 'moving' the continents he could reverse the slope of the $\delta^{18}O$ -temperature curve. Experiments like this raise the question of how much chemistry and physics there is in regression fitting. The authors should comment on this here.

Line 212 (and elsewhere): 'blank' is perhaps a misleading word to use here. 'snow-free' is perhaps a less ambiguous expression.

Line 258-260: An important point and finding here.

Line 280: Many readers outside Germany may be unfamiliar with the PANGAEA web site. Please present this here, namely:

www.pangaea.de/

line 283: Authors should present rationale for this derivation. Define it this way it is easy to show that $q_{sea} = q_{air}$. As such, using the bulk aerodynamic formula, there would be no evaporation! Please justify this estimation.

Line 305: Why was VSMOW2 not used as the reference (Tyler B. Coplen, 2011: Guidelines and recommended terms for expression of stable-isotope-ratio and gas-ratio measurement results. *Rapid Communications in Mass Spectrometry*, 25, 2538-2560, doi: 10.1002/rcm.5129).

References

Many of the reference details are incomplete which would make hard the task for the reader to find the relevant papers. Some I noticed were:

Sodemann, H., V. Masson-Delmotte, C. Schwierz, B. M. Vinther and H. Wernli, 2008: Interannual variability of Greenland winter precipitation sources: 2. Effects of North Atlantic Oscillation variability on stable isotopes in precipitation. *J. Geophys. Res.*, 113, D12111, doi: 10.1029/2007JD009416.

Steen-Larsen, H. C., et al. (2015), Moisture sources and synoptic to seasonal variability of North Atlantic water vapor isotopic composition, *J. Geophys. Res.*, 120, 5757-5774, doi: 10.1002/2015JD023234.

Naoyuki Kurita, 2011: Origin of Arctic water vapor during the ice-growth season. *Geophysical Research Letters*, 38, L02709, doi: 10.1029/2010GL046064.

Eric S. Klein, and Jeffrey M. Welker, 2016: Influence of sea ice on ocean water vapor isotopes and Greenland ice core records. *Geophysical Research Letters*, 43, 12475-12483, doi: 10.1002/2016GL071748.

Kurita, N., et al. (2016), Influence of large-scale atmospheric circulation on marine air intrusion toward the East Antarctic coast, *Geophys. Res. Lett.*, 43, 9298-9305, doi: 10.1002/2016GL070246.

Werner M, Langebroek PM, Carlsen T, Herold M, Lohmann G (2011) Stable water isotopes in the ECHAM5 general circulation model: Toward high-resolution isotope modeling on a global scale. *J. Geophys. Res.* 116: D15109 doi: 10.1029/2011JD015681.

Reviewer #2 (Remarks to the Author):

Review: Water vapour isotopic composition over open ocean and sea ice in the Atlantic sector by Bonne et al.

This manuscript presents, and thoroughly, analyses a new set of observations conducted from the Polarstern research ship over a period of two years. As it sailed from pole-to-pole, within the Atlantic sector, the authors and their larger team, took continuous high quality measurements of stable water isotopes both in seawater and in vapour above the ship.

The focus of the manuscript is on controls over boundary layer water vapour $d_{18}O$ and d -excess. These controls are assessed using the large new Polarstern dataset. The investigation of the impacts of sea ice and wind speed on d -excess are particularly important. I find the investigation to be a thorough and well written piece of analysis which is entirely suitable for publication in Nature Comms. The research is statistically sound, and will be highly influential within the field. All my comments are of a minor nature: mostly they are suggestions to improve the clarity of the figures or writing.

Detailed comments:

Sometimes sea-ice, sometimes sea ice.

Abstract: The abstract is good. And I agree with the authors that this is a very important new set of vapour observations that allows new insights into key isotopic processes. On the last sentence, it would be useful if the authors could spell out more clearly some of the findings will be used to 'enhance the interpretation of water isotopes in paleo-climate archives'.

L30-31: Slightly awkward phrasing for this sentence.

L71: Explain first and second order isotopic signal

The dataset is mostly well described.

L102: where/were

L108: "very well reproduces" awkward phrasing.

L108-113: How was the simulation run? Was it nudged?

L144:145 and Figure 1. It is really difficult to see the obs versus modelled results. And the way the legend is formatted makes this difficult to read too. A better way of plotting these data would be useful.

L237...252 or so: Do results need to be repeated here? It feels unnecessary.

L252-270 Much more useful. Cite Holloway Sime et al 2016 under the sea ice change part?

L293-300 : Why not use NSIDC sea ice data?

Figure 1: very confusing legend and inability to see modelled versus obs values.

Figure 2: Why is this data not also shown on Figure 1 for comparison?

Figure 3: Nice, but maybe better to use different colours for the fits? The fitted lines and dashed lines disappear into the data points using the current scheme.

Supplementary information: Bare. Not blank.

L.C.Sime

Reviewer #3 (Remarks to the Author):

The paper presents a unique, highly interesting data set of stable isotope ratios of near-surface water vapor and ocean surface water obtained from the research vessel Polarstern on several transects between the Arctic Ocean and the Southern Ocean. Measurements were carried out over a two-year time period. The data are used to evaluate the traditional model by Merlivat and Jouzel (1979) as well as isotope-enabled GCMs, the latter based on the MJ79 model. The authors state that the isotopic composition of near-surface water vapor above the ocean is highly variable. They conclude that generally the d-excess depends on a combination of SST and relative humidity, whereas the wind speed influence postulated by Merlivat and Jouzel could not be confirmed. At high latitudes, the influence of sea ice was investigated, and it was found that the snow cover of the sea ice has the largest influence on d-excess of the vapor since its isotopic signal is clearly distinguishable from that of ocean water.

General comments:

The paper is well written, the English is ok (if not flawless, though), and the methods are basically sound. My main concern is the conclusion that the wind speed does not play any role for the d-excess of the vapor evaporated from the ocean (or snow, at high latitudes). I agree that the treatment of wind speed in MJ79 using a smooth and rough regime with some seemingly arbitrary threshold (and a quite low upper limit for wind speed anyway) is not ideal. The pioneering work of Jouzel and Merlivat is highly appreciated, however, I think it is high time that we stop evaluating their model using the abundance of modern data, but try to understand the involved physical processes independently. The authors do not give a physical explanation for the unimportance of wind speed. The conclusion is actually only that the wind speed as treated in MJ79 (and in GCMs) does not have a large influence on the d-excess. To really investigate the influence of the wind speed it would be necessary to study the relationship between wind speed and deuterium excess for otherwise constant conditions. Fig. 4 only shows d-excess in dependence of RH for the smooth and rough regime, however, it does not show the actual wind speed, and, additionally, a certain relative humidity does not mean constant conditions, since it depends on the temperature, and higher temperatures mean higher vapor fluxes, which influences the fractionation. It would be interesting to see a plot wind speed vs. d-excess for a series of constant T-RH combinations. The presented data set would be very valuable to investigate this (which was not possible at the time Jouzel and Merlivat did their studies due to lack of data). I would be really interested in seeing such an analysis and it would considerably increase the importance of their paper. Without this suggested analysis, the authors should be very careful with the formulation of their conclusion and restrict it to treatment of wind speed as in the models.

Specific comments:

L15:moisture for climate.

L27: replace "enhance" by improve

L33: "climate-related" studies would be enough

L34: better use "precipitation" rather than rainfall

L49: the latter

L65: Atlantic Ocean

L94: delete "of"

L98: transporting isotopically depleted water originating from large Siberian rivers southward along the eastern Greenland coast

L102: Atlantic Ocean

L119: what does 137% rel. humidity mean??

L134ff: see general comments

L190: 2% seems to be a very low threshold, and the plot it looks

like sea-ice concentrations were usually way above 2% if sea ice were there at all. Evaporation from the ocean already plays a significant role for sea ice concentrations smaller than 90%. Please explain the choice of 2% as a threshold.

L197: substantially

L212: better use "bare" or "snow-free" rather than blank

L310: on board of the research vessel Polarstern

L375: 60m above the surface

L385: on top of the sea ice

L393: free of both...

L395: ...the isotope data.. are selected... by including al measurements...

L503: data of this study are available...

Fig. 1 very hard to read since model and observation are relatively close together

x-axis: strange choice of labels...

Suppl. Info

L12: It does, however, depend..

L29: check formulation “ distant from this running average from...”

L59: locations for which...

L133: blank, see above

Answers to the reviewers

“Water vapour isotopic composition over open ocean and sea ice in the Atlantic sector”

Reviewer #1 (Remarks to the Author):

Increasingly stable water isotopes are being seen as a key component of analysing many aspects of the workings of the global circulation, and particularly the transport of water vapor. The focus here is on deuterium-excess and what it reveals about the local boundary layer over the ocean. The authors posit that this excess is controlled by the relative humidity and ocean temperature, and not by wind speed. They argue that the various influences (and their relative importance) on d-excess have in the past only been determined in atypical locations such as coastal Iceland and Bermuda.

The authors present new dataset of in situ isotopic measurements of ocean surface water and vapor which was obtained from an extensive Polarstern cruise covering the global Atlantic sector. Data on the cruise were collected every 6 hr over the period mid-2015 to mid-2017.

This manuscript is potentially a valuable contribution to this understanding. However, I outline below some aspect in which the submission should be revised.

Lines 79-81: Figure 1 presents a neat overview of the observations and the location (latitude) of Polarstern. The authors should make clear how the ‘average values’ were calculated. If this was done with the 6-hr data we would not get a true picture of the Atlantic average, as the ship spends most time in the polar regions, and relatively little time equatorward of 45 degrees (as deduced from bottom panel of Fig.). Please clarify. As a broader comment, I am not sure whether these average values have much meaning.

The average values have been simply calculated based on the 6 hours values of the whole dataset. We agree that this information is not absolutely necessary for the argumentation but was simply given to provide the reader a first idea of the dataset. These average values have been removed from the manuscript. The description of minima and maxima which follows is to our opinion a more pertinent piece of information.

Also, in the Fig. it is hard to differentiate between the ECHAMexp and ECHAMfinal curves. Can you make this easier for the reader (e.g., increase the contrast, use dash lines etc.)

The figure has been updated. We hope the new version is more easily readable.

Line 61: Make reference here also to the analysis of Noone et al., 2004: Sea ice control of water isotope transport to Antarctica and implications for ice core interpretation. *J. Geophys. Res.*, 109, D07105, doi:10.1029/2003JD004228.

The reference has been added.

Line 91 (and throughout): It would be appropriate to refer to the 79 deg glacier as Nioghalvfjærdsbrae. Corrected in the text and figures.

Lines 94-96: This statement is vague and misleading. Perhaps refer to one of the standard global climatologies (e.g., the GISS product) which show peak values of oceanic $\delta O18$ at about 20degrees latitude (just equatorward of the subtropical ridge in both hemispheres. The key factor here is P-E, not evaporation.

This paragraph has been reformulated. The key parameter, precipitation-evaporation is now specifically cited instead of referring to only evaporation. The reference to the GISS database, with which the comparison to our dataset gives coherent results, has also been included.

Lines 115-127: A problem I have with the discussion of the correlations of the ‘baseline’ values is that the data are varying in both space and time. If two variables are correlated in the space domain this, a priori, tells us nothing about how they might be correlated in time. A spatial relationship between two variables in general may say nothing about the temporal relationship. The classic Dansgaard 1964

study showed a strong annual SPATIAL relationship between T and $\delta\text{O}18$, the SEASONAL cycle of parameters often yields associations opposite to those of the spatial analysis. For example, $\delta\text{O}18$ in summer precipitation at Lhasa (Tibetan Plateau) is depleted by more than 6 ppt with respect to winter rainfall, despite the surface air temperature being 10°C higher in summer. (This is because gradual rain-out of moist, oceanic air masses moving inland, associated with the monsoon circulation, constitutes a powerful mechanism capable of producing large isotopic depletions in rainfall, often completely overshadowing the dependence of them on temperature.) Please word this part of the text more carefully.

We completely agree with the reviewer on this topic. Spatial and temporal relationships between isotope data and any climate variable (e.g. temperature, precipitation amount) should never be assumed to be equal, a priori. The same is true for relationships on different temporal scales, e.g. seasonal changes, interannual changes or glacial-interglacial changes. We have rephrased the text, accordingly.

In a similar vein the regressions shown in Figure 3 reflect a mixture of time and space sampling, and the authors should make clearer the physical argument they are presenting. I am reminded of the study of Brown et al., 2004: The dependence of the $\delta\text{O}18$ -temperature relationship on continent locations. *Geophys. Res. Lett.*, 31, L09208 who conducted experiments with an isotope-enabled GCM. He found that by ‘moving’ the continents he could reverse the slope of the $\delta\text{O}18$ -temperature curve. Experiments like this raise the question of how much chemistry and physics there is in regression fitting. The authors should comment on this here.

Again, we fully agree with the reviewer. A regression fit does not give a physical (or chemical) explanation for the observed isotope changes. However, it is not the goal of this part of the study to give such physical explanation, but to simply present an empirical link between Deuterium excess, SST and RH for our extensive data set. We compare our data to physical-based values of the MJ79 model and the isoGCM results later in the manuscript.

As the data set covers a much wider range of different meteorological conditions as compared to previous studies, we rate our empirical function as valuable for the scientific community. We also think that it does have a general relevance, as the studied processes occurring at the ocean surface are strongly dominated by local evaporation. However, further measurements (e.g. in the Pacific realm) are required to test the validity of our empirical function for different spatial and temporal domains. This limitation is now explicitly stated in the revised manuscript.

Line 212 (and elsewhere): ‘blank’ is perhaps a misleading word to use here. ‘snow-free’ is perhaps a less ambiguous expression.

We decided to use the word “bare” sea ice, instead of “blank” as suggested by the other reviewers and corrected this expression in the whole manuscript.

Line 258-260: An important point and finding here.

Line 280: Many readers outside Germany may be unfamiliar with the PANGAEA web site. Please present this here, namely:

www.pangaea.de/

The direct link to the dataset should have been given in the references, as it has a doi. However, there had been a problem with the citations, and the numbers were not pointing to the right reference. This has been corrected.

Line 283: Authors should present rationale for this derivation. Define it this way it is easy to show that $q_{\text{sea}} = q_{\text{air}}$. As such, using the bulk aerodynamic formula, there would be no evaporation! Please justify this estimation.

This is the same formulation as used by Benetti et al., 2014 or Benetti et al., 2017. Therefore, we used it for consistency and comparability with these studies focusing on similar questions and observing systems.

As no direct measurement of the relative humidity were performed at the sea surface, we use this formula to calculate RH. We don't consider it as the real value of RH_{sea} , but the best estimate we can get from our relative humidity measurements in the air at 29 m altitude (RH_{air}). We have modified the text to explain this approximation in more detail.

Line 305: Why was VSMOW2 not used as the reference (Tyler B. Coplen, 2011: Guidelines and recommended terms for expression of stable-isotope-ratio and gas-ratio measurement results. Rapid Communications in Mass Spectrometry, 25, 2538-2560, doi: 10.1002/rcm.5129).

The calibration standards and oceanic samples have been measured with an instrument calibrated on the VSMOW-SLAP scale. Therefore, we use this reference in the manuscript. The very limited difference on δ^2H between VSMOW-SLAP and VSMOW2-SLAP2 scales would not affect our results significantly.

However, according to the recommendations from Coplen 2011, we decided to replace the notation δD by δ^2H throughout the manuscript.

References

Many of the reference details are incomplete which would make hard the task for the reader to find the relevant papers. Some I noticed were:

Sodemann, H., V. Masson-Delmotte, C. Schwierz, B. M. Vinther and H. Wernli, 2008: Interannual variability of Greenland winter precipitation sources: 2. Effects of North Atlantic Oscillation variability on stable isotopes in precipitation. J. Geophys. Res., 113, D12111, doi: 10.1029/2007JD009416.

Steen-Larsen, H. C., et al. (2015), Moisture sources and synoptic to seasonal variability of North Atlantic water vapor isotopic composition, J. Geophys. Res., 120, 5757-5774, doi: 10.1002/2015JD023234.

Naoyuki Kurita, 2011: Origin of Arctic water vapor during the ice-growth season. Geophysical Research Letters, 38, L02709, doi: 10.1029/2010GL046064.

Eric S. Klein, and Jeffrey M. Welker, 2016: Influence of sea ice on ocean water vapor isotopes and Greenland ice core records. Geophysical Research Letters, 43, 12475-12483, doi: 10.1002/2016GL071748.

Kurita, N., et al. (2016), Influence of large-scale atmospheric circulation on marine air intrusion toward the East Antarctic coast, Geophys. Res. Lett., 43, 9298-9305, doi: 10.1002/2016GL070246.

Werner M, Langebroek PM, Carlsen T, Herold M, Lohmann G (2011) Stable water isotopes in the ECHAM5 general circulation model: Toward high-resolution isotope modeling on a global scale. J. Geophys. Res. 116: D15109 doi: 10.1029/2011JD015681.

The reference has been list has been created using the standard Nature referencing style from the Zotero tool. We believe it fits the recommendations from the journal.

Reviewer #2 (Remarks to the Author):

This manuscript presents, and thoroughly, analyses a new set of observations conducted from the Polarstern research ship over a period of two years. As it sailed from pole-to-pole, within the Atlantic sector, the authors and their larger team, took continuous high quality measurements of stable water isotopes both in seawater and in vapour above the ship.

The focus of the manuscript is on controls over boundary layer water vapour d18O and d-excess. These controls are assessed using the large new Polarstern dataset. The investigation of the impacts of sea ice and wind speed on d-excess are particularly important. I find the investigation to be a thorough

and well written piece of analysis which is entirely suitable for publication in Nature Comms. The research is statistically sound, and will be highly influential within the field. All my comments are of a minor nature: mostly they are suggestions to improve the clarity of the figures or writing.

Detailed comments:

Sometimes sea-ice, sometimes sea ice.

We corrected to sea ice without hyphen in the whole manuscript.

Abstract: The abstract is good. And I agree with the authors that this is a very important new set of vapour observations that allows new insights into key isotopic processes. On the last sentence, it would be useful if the authors could spell out more clearly some of the findings will be used to 'enhance the interpretation of water isotopes in paleo-climate archives'.

Paleoclimate studies can benefit from our results for the interpretation of changes related to moisture source or sea ice extent variations. The last sentence of the abstract has been modified.

L30-31: Slightly awkward phrasing for this sentence.

The sentence has been modified.

L71: Explain first and second order isotopic signal

Isotopic abundances of different species are the first order signal, and the d-excess, which is based on the first order signal of $\delta^{18}O$ and δ^2H , is the second order signal. This has been added in the text

The dataset is mostly well described.

L102: where/were

Corrected

L108: "very well reproduces" awkward phrasing.

Sentence modified: "A simulation with an isoGCM reproduces very well the atmospheric isotope measurements"

L108-113: How was the simulation run? Was it nudged?

Yes, the simulation was nudged to meteorology (to ERA-interim surface pressure, temperature, vorticity and divergence fields), as described in the Methods section. For more clarity, we added in the main text that the simulation was nudged.

L144:145 and Figure 1. It is really difficult to see the obs versus modelled results. And the way the legend is formatted makes this difficult to read too. A better way of plotting these data would be useful.

The figure has been modified in order to make it more straightforward for the reader.

L237...252 or so: Do results need to be repeated here? It feels unnecessary.

We believe it is useful to summarize the main results in the conclusion of the paper. This part has however been shortened to avoid a long repetition of already detailed explanation.

L252-270 Much more useful. Cite Holloway Sime et al 2016 under the sea ice change part?

A reference to this study has been included.

L293-300 : Why not use NSIDC sea ice data?

We have compared the ERA-interim reanalysis data with the satellite products ASMR2 and found a very good coherence. As the ECHAM5-wiso model is also nudged to the ERA reanalyses, using this dataset for the interpretation of our observation was also coherent.

Figure 1: very confusing legend and inability to see modelled versus obs values.
The figure has been improved.

Figure 2: Why is this data not also shown on Figure 1 for comparison?
The oceanic dataset has been integrated as time series in Figure 1.

Figure 3: Nice, but maybe better to use different colours for the fits? The fitted lines and dashed lines disappear into the data points using the current scheme.
We believe the colours of the observed scattered dataset and the lines should be the same, as it allows an easy comparison of the fit and the closure assumption values with the observations. In order to distinguish the lines better from the observations, we however decided to increase the line width and to draw edges around the scatters.

Supplementary information:

Bare. Not blank.

We replaced “blank” sea ice by “bare” sea ice in the whole manuscript.

Reviewer #3 (Remarks to the Author):

The paper presents a unique, highly interesting data set of stable isotope ratios of near-surface water vapor and ocean surface water obtained from the research vessel Polarstern on several transects between the Arctic Ocean and the Southern Ocean. Measurements were carried out over a two-year time period. The data are used to evaluate the traditional model by Merlivat and Jouzel (1979) as well as isotope-enabled GCMs, the latter based on the MJ79 model. The authors state that the isotopic composition of near-surface water vapor above the ocean is highly variable. They conclude that generally the d-excess depends on a combination of SST and relative humidity, whereas the wind speed influence postulated by Merlivat and Jouzel could not be confirmed. At high latitudes, the influence of sea ice was investigated, and it was found that the snow cover of the sea ice has the largest influence on d-excess of the vapor since its isotopic signal is clearly distinguishable from that of ocean water.

General comments:

The paper is well written, the English is ok (if not flawless, though), and the methods are basically sound. My main concern is the conclusion that the wind speed does not play any role for the d-excess of the vapor evaporated from the ocean (or snow, at high latitudes). I agree that the treatment of wind speed in MJ79 using a smooth and rough regime with some seemingly arbitrary threshold (and a quite low upper limit for wind speed anyway) is not ideal. The pioneering work of Jouzel and Merlivat is highly appreciated, however, I think it is high time that we stop evaluating their model using the abundance of modern data, but try to understand the involved physical processes independently. The authors do not give a physical explanation for the unimportance of wind speed. The conclusion is actually only that the wind speed as treated in MJ79 (and in GCMs) does not have a large influence on the d-excess. To really investigate the influence of the wind speed it would be necessary to study the relationship between wind speed and deuterium excess for otherwise constant conditions. Fig. 4 only shows d-excess in dependence of RH for the smooth and rough regime, however, it does not show the actual wind speed, and, additionally, a certain relative humidity does not mean constant conditions, since it depends on the temperature, and higher temperatures mean higher vapor fluxes, which influences the fractionation. It would be interesting to see a plot wind speed vs. d-excess for a series of constant T-RH combinations. The presented data set would be very valuable to investigate this (which was not possible at the time Jouzel and Merlivat did their studies due to lack of data). I would be really interested in seeing such an analysis and it would considerably increase the importance of their paper. Without this suggested analysis, the authors should be very careful with the

formulation of their conclusion and restrict it to treatment of wind speed as in the models.

We also really appreciate the theory of Merlivat and Jouzel. The related closure assumption can easily provide a first estimation of the oceanic boundary layer water vapour isotopic composition. Using the MJ79 approach, we could simulate most of the d-excess signal and explain its relation to relative humidity and sea surface temperature due to the kinetic fractionation processes occurring during evaporation.

As suggested by the reviewer, we performed a complementary analysis of d-excess as a function of wind speed under specific conditions of RH_{sea} and SST. For this purpose, we split our dataset into subsets of different ranges of SST and RH_{sea} . The different ranges of SST and RH_{sea} are chosen in such a way that variations of each of these parameters are small, but that at the same time enough data points for a robust analysis are contained in each subset.

The results, presented in Figure 1 below, do not show any particular relationship between the wind speed and the d-excess for any combination of RH_{sea} and SST range. From this analysis we cannot exclude that such relationship might exist at the microphysical scale, as demonstrated in laboratory experiments by Merlivat and Jouzel. However, our data set reveals that such wind speed dependency cannot be detected on a larger spatial scale in the atmosphere boundary layer vapour d-excess signal over the Atlantic Ocean.

Figure 1: Observed d-excess as a function of wind speed (blue dots) for different ranges of RH_{sea} and SST. All horizontally (vertically) distributed subplots have the same ranges of SST values (RH_{sea} values). From left to right, RH_{sea} ranges are successively: 50 - 60%, 60 - 70%, 70 - 80%, 80 - 90%, 90 - 100% and 100 - 125%. From bottom to top, SST ranges are: 1.8 - 5°C, 5 - 10°C, 10 - 15°C, 15 - 20°C, 20 - 25°C and 25 - 30°C. Vertical red lines represent the 7 m/s wind speed limit.

Specific comments:

L15:moisture for climate.

Corrected

L27: replace “enhance” by improve

Corrected. Also changed “improved” by “better” on line 26 to avoid repetition.

L33: “climate-related” studies would be enough

Corrected

L34: better use “precipitation” rather than rainfall

Corrected

L49: the latter

Corrected

L65: Atlantic Ocean

Our dataset is mostly spread around the Atlantic Ocean, but also includes periods in the Arctic and Southern Ocean. For this reason, we had chosen the term Atlantic sector, which is less precise but more appropriate for the regional extent of our dataset.

L94: delete “of”

Corrected.

L98: transporting isotopically depleted water originating from large Siberian rivers southward along the eastern Greenland coast

Corrected.

L102: Atlantic Ocean

See comment for L65.

L119: what does 137% rel. humidity mean??

This value corresponds to only one single isolated data point, which is rated as an outlier. It has been removed from the analyses.

L134ff: see general comments

See the additional evaluation of the separate influences of wind speed in distinct SST and RH_{sea} domains.

L190: 2% seems to be a very low threshold, and the plot it looks like sea-ice concentrations were usually way above 2% if sea ice were there at all. Evaporation from the ocean already plays a significant role for sea ice concentrations smaller than 90%. Please explain the choice of 2% as a threshold.

The 2% threshold was originally chosen instead of 0% to avoid considering the presence of small sea ice variations due to signal noise when sea ice data from satellite products AMSR2² (Spren et al. 2008) was used in some first, preliminary analyses. This criterion is obsolete for the final analyses based on the ERA sea ice dataset. We now use 0% as the limit and updated the results accordingly. This small modification of the sea ice limit leads to negligible changes of our results, only.

L197: substantially

Corrected.

L212: better use “bare” or “snow-free” rather than blank
Ok. We corrected this wording in the whole manuscript.

L310: on board of the research vessel Polarstern
Corrected.

L375: 60m above the surface
Corrected.

L385: on top of the sea ice
Corrected.

L393: free of both...
Corrected.

L395: ...the isotope data.. are selected... by including al measurements...
Corrected.

L503: data of this study are available...
Corrected

Fig. 1 very hard to read since model and observation are relatively close together
x-axis: strange choice of labels...

The colours of the different datasets have been modified to more distinguishable from each other. The new labels should be more readable, too.

Suppl. Info

L12: It does, however, depend..
Corrected

L29: check formulation “distant from this running average from...”
Corrected. New formulation: “... eliminate any measurements whose distance from this running average exceeds the thresholds of...”

L59: locations for which...
Corrected

L133: blank, see above
Corrected

References:

1. Meyer, H., Schönicke, L., Wand, U., Hubberten, H. W. & Friedrichsen, H. Isotope Studies of Hydrogen and Oxygen in Ground Ice - Experiences with the Equilibration Technique. *Isotopes in Environmental and Health Studies* **36**, 133–149 (2000).
2. Spreen, G., Kaleschke, L. & Heygster, G. Sea ice remote sensing using AMSR-E 89-GHz channels.

Journal of Geophysical Research: Oceans **113**,

Reviewer #1 (Remarks to the Author):

The authors have addressed appropriately all the issues I raised in my review of the original submission.

I am pleased to now be able to recommend acceptance

Reviewer #2 (Remarks to the Author):

I am happy enough with the authors responses to the first round of reviewers comments. They seem to have done a satisfactory job of responding to all comments.

Reviewer #3 (Remarks to the Author):

General remarks:

The authors did a thorough revision of the manuscript and addressed all points the reviewers had in their "Answers to the reviewers".

However, particularly in my case, they did not include the main points in the paper. If readers have the same comments as I had, they do not get the information the authors gave to the reviewer, which is not very satisfactory.

Specific comments:

Concerning the influence on wind speed on the deuterium excess:

I think generally you should make clear whether you are talking about testing the MJ79 findings or whether you are talking about general physics.

It would be good to give a physical explanation why the wind speed should influence d-excess according to MJ1979, and also possible explanations why it should not according to your findings.

The interesting figure you showed to the reviewers should be included in the supplementary material and should be discussed in the text so that the reader can comprehend your statement. It is an important and valuable finding and you should not hide this from the reader.

From your investigation with the smooth and rough regime alone you can only state that not all findings of MJ79 are confirmed, but not that there is generally no dependence of d-excess on wind speed.(L144-146)

I also have an additional point (I apologize for not having mentioned it in my first review):

You investigate the influence of sea ice on the stable isotope ratios and in the discussion of Greenland you nicely mention the possibility of “vapour blown from the Greenland ice sheet towards the research vessel”. However, this is the only case you include the influence of advection in your analysis. Antarctic sea ice is mainly found in the vicinity of the continent, which is strongly influenced by synoptic activity in the circumpolar trough, and close to the continent, can also be influenced by katabatic outflow. You never mention anything like that. You assume that the water vapour isotopology depends solely on local evaporation/sublimation conditions. Is that a valid assumption? I think it is not. It is most likely beyond the scope of the presented study to include an analysis of advection, however, these things should be mentioned in the discussion.

I also tried to figure out from Suppl. Fig. 3 whether increasing sea ice also generally meant increasing latitude (at least in Antarctica) (thus decreasing temperature, decreasing $\delta^{18}O$ etc.), but it was impossible to get this information from the tiny figure that, particularly for Antarctica, has the sea ice information in a stretch hardly bigger than 1cm. I am also not sure if it makes sense to compare the entire open water data, which include the tropics and sub-tropics, to the sea ice data since those are completely different climatological regimes.

Using d-excess from ice cores alone as a proxy for sea ice seems to be a slightly bold approach. It could be combined with the analysis of other ice core properties, namely chemical ones, to yield a more accurate estimate of former sea ice extents.

Small comments:

L65: Atlantic sector:

Since there are different definitions for the oceans (e.g. the Southern Ocean is also defined as the southernmost parts of the Atlantic, Pacific and Indian Ocean) it would be good to just give a short

explanation of what you mean by “Atlantic sector”. (For me it was a bit confusing, since “sector” used to be often used for parts of the Antarctic continent). (similarly “Antarctic sector”)

L99: “In some the Arctic region”

Is that supposed to mean “in some parts” or “in the Arctic region”?

Answers to the reviewers

“Water vapour isotopic composition over open ocean and sea ice in the Atlantic sector”

We thank reviewers #1 and #2 for their positive answers and recommendation of acceptance. We also appreciate the comments of reviewer #3, which are again very helpful for improving this manuscript.

Reviewer #3 (Remarks to the Author):

General remarks:

The authors did a thorough revision of the manuscript and addressed all points the reviewers had in their “Answers to the reviewers”.

However, particularly in my case, they did not include the main points in the paper. If readers have the same comments as I had, they do not get the information the authors gave to the reviewer, which is not very satisfactory.

In case of publication, we intend to opt in for the Nature Communication scheme to publish all communication with the reviewers together with the article. Thus, the readers will have the same information as the reviewers.

Specific comments:

Concerning the influence on wind speed on the deuterium excess:

I think generally you should make clear whether you are talking about testing the MJ79 findings or whether you are talking about general physics.

It would be good to give a physical explanation why the wind speed should influence d-excess according to MJ1979, and also possible explanations why it should not according to your findings. The interesting figure you showed to the reviewers should be included in the supplementary material and should be discussed in the text so that the reader can comprehend your statement. It is an important and valuable finding and you should not hide this from the reader.

From your investigation with the smooth and rough regime alone you can only state that not all findings of MJ79 are confirmed, but not that there is generally no dependence of d-excess on wind speed. (L144-146)

Merlivat and Jouzel (1979) derived a theoretical model to account for the deuterium-oxygen 18 relationship measured in meteoric waters. For the processes occurring during the evaporation of water, they have used results from a model by Brutsaert (1975). According to MJ79, this model was shown to be in good agreement with observed fractionation of the isotopic species of water occurring when water is evaporated and transported into the atmosphere under various laboratory (wind tunnel) conditions. Following the model of Brutsaert, in MJ79 a distinction between smooth and rough water-atmosphere interfaces is proposed. The domains of validity of these two interfaces are experimentally defined. A jump between both interfaces is proposed for surface roughness Reynolds numbers below or above 1.

As the content of this paper is based on experimental data taken within the atmospheric boundary layer at 29 m height above sea level, we do not pretend to be able to evaluate the physics of the MJ79 model, as our measurements do not capture the microphysical scale of evaporation processes happening within the skin layer at the ocean-air interface. However, our measurements clearly reveal that - at a larger spatial scale - isotopic vapour data within the atmospheric boundary layer is not conform with the MJ79 model. Furthermore, we demonstrate that both simple isotope modeling (i.e. calculations based on the closure assumption) as well as more complex modelling approaches (isotopes within the atmosphere GCM ECHAM5-wiso) match better to the observations if the distinction between smooth and rough evaporative regimes is discarded.

We have rephrased parts of the manuscript to clarify this difference between the MJ79 model approach and our own findings.

Regarding the figure shown in our previous reply: It was never our intention to hide anything from the reader and we have now included the figure in the supplementary material, as suggested (new Supplementary Figure 4). We have added a discussion on this new figure in the text.

I also have an additional point (I apologize for not having mentioned it in my first review): You investigate the influence of sea ice on the stable isotope ratios and in the discussion of Greenland you nicely mention the possibility of “vapour blown from the Greenland ice sheet towards the research vessel”. However, this is the only case you include the influence of advection in your analysis. Antarctic sea ice is mainly found in the vicinity of the continent, which is strongly influenced by synoptic activity in the circumpolar trough, and close to the continent, can also be influenced by katabatic outflow. You never mention anything like that. You assume that the water vapour isotopology depends solely on local evaporation/sublimation conditions. Is that a valid assumption? I think it is not. It is most likely beyond the scope of the presented study to include an analysis of advection, however, these things should be mentioned in the discussion.

To evaluate the potential influence of an advection of air originating from the continent on our dataset used for the sea ice analyses, we applied an additional filter considering the wind direction and speed, as it was already done for the dataset representative of the open ocean. We removed from the sea ice dataset the periods where the observed wind was originating from a coastal region. Only a limited number of data points are removed by this additional filter (67 out of a total of 843 data points), as most of the areas of observations which are covered by sea ice are situated far from the coasts. As it can be seen in Figure A, such filtering does not affect the relationship between the sea-ice cover and the observed d-excess significantly. This information has been included in the manuscript.

Figure A: Sensitivity of the deuterium excess in vapour with respect to different sea ice coverage conditions. (a) Same as Manuscript Figure 5, (b) with an additional filter removing any data points potentially influenced by wind originating from land-covered regions.

I also tried to figure out from Suppl. Fig. 3 whether increasing sea ice also generally meant increasing latitude (at least in Antarctica) (thus decreasing temperature, decreasing $\delta^{18}O$ etc.), but it was impossible to get this information from the tiny figure that, particularly for Antarctica, has the sea ice information in a stretch hardly bigger than 1cm.

Around Antarctica, the sea ice cover is generally increasing when going southward. However, the latitudinal ship position and the sea ice cover surrounding the research vessel are not strongly correlated during our observational period (Fig. B). This might be explained by the fact that the ship usually tries avoiding the most compact ice regions to secure navigation.

As pointed out by the reviewer, the sea ice cover map around Antarctica from Supplementary Figure 3.b was lacking of readability. Therefore, we improved this figure by zooming-in on smaller regions covered by sea ice.

Figure B: Sea ice cover (SIC) surrounding Polarstern, as a function of the latitude for the Southern Hemisphere only. The color indicates the observed air temperature.

I am also not sure if it makes sense to compare the entire open water data, which include the tropics and sub-tropics, to the sea ice data since those are completely different climatological regimes.

As stated in the Method section in the paragraph “Data filtering”, for the open-water related analyses we have excluded any data points potentially influenced by sea ice coverage. Vice versa, the sea ice related analyses only include data points where sea ice surrounded the Polarstern. Thus, we believe that the interpretations of observations at open sea and in sea ice covered areas are treated in an adequate independent manner in this study.

Using d-excess from ice cores alone as a proxy for sea ice seems to be a slightly bold approach. It could be combined with the analysis of other ice core properties, namely chemical ones, to yield a more accurate estimate of former sea ice extents.

The d-excess should indeed be used as a complement with other sea-ice proxies. This has been added in the manuscript.

Small comments:

L65: Atlantic sector:

Since there are different definitions for the oceans (e.g. the Southern Ocean is also defined as the southernmost parts of the Atlantic, Pacific and Indian Ocean) it would be good to just give a short explanation of what you mean by “Atlantic sector”. (For me it was a bit confusing, since “sector” used to be often used for parts of the Antarctic continent). (similarly “Antarctic sector”)

An explanation has been added: Atlantic sector is considered as the Atlantic Ocean and the Atlantic regions of the Arctic and Southern Oceans.

L99: “In some the Arctic region”

Is that supposed to mean “in some parts” or “in the Arctic region”?

Corrected: „In some parts of the Arctic“

Reviewer #3 (Remarks to the Author):

3rd review of Bonne et al., submitted to Nature Comm.

I marked my new comments with "R3", since my formatting got lost when I pasted my review into this field.

Generally I still think that this is a highly interesting paper, dealing with a unique data set, I only think that some aspects, especially simplifying assumptions, should be explained better and some results should be formulated more cautiously.

I will comment on the response to my latest review. My old response in normal font, the answer of the authors in italic, my present response in bold font.)

General remarks:

The authors did a thorough revision of the manuscript and addressed all points the reviewers had in their "Answers to the reviewers".

However, particularly in my case, they did not include the main points in the paper. If readers have the same comments as I had, they do not get the information the authors gave to the reviewer, which is not very satisfactory.

In case of publication, we intend to opt in for the Nature Communication scheme to publish all communication with the reviewers together with the article. Thus, the readers will have the same information as the reviewers.

R3:A publication should be self-contained and not many people have the time to read all the papers they want to read, let alone additionally all the correspondence with the reviewers of the papers. However, I leave it to the editor to decide whether this is the desired way to handle information. (From my point of view, even the supplementary information should not be necessary to basically understand the paper, but only for people with a deeper interest in the subject.)

Specific comments:

Concerning the influence on wind speed on the deuterium excess:

I think generally you should make clear whether you are talking about testing the MJ79 findings or whether you are talking about general physics.

It would be good to give a physical explanation why the wind speed should influence d-excess according to MJ1979, and also possible explanations why it should not according to your findings. The interesting figure you showed to the reviewers should be included in the supplementary material and should be discussed in the text so that the reader can comprehend your statement. It is an important and valuable finding and you should not hide this from the reader. From your investigation with the smooth and rough regime alone you can only state that not all findings of MJ79 are confirmed, but not that there is generally no dependence of d-excess on wind speed. (L144-146)

Merlivat and Jouzel (1979) derived a theoretical model to account for the deuterium-oxygen 18 relationship measured in meteoric waters. For the processes occurring during the evaporation of water, they have used results from a model by Brutsaert (1975). According to MJ79, this model was shown to be in good agreement with observed fractionation of the isotopic species of water occurring when water is evaporated and transported into the atmosphere under various laboratory (wind tunnel) conditions. Following the model of Brutsaert, in MJ79 a distinction between smooth and rough water- atmosphere interfaces is proposed. The domains of validity of these two interfaces are experimentally defined. A jump between both interfaces is proposed for surface roughness Reynolds numbers below or above 1.

As the content of this paper is based on experimental data taken within the atmospheric boundary layer at 29 m height above sea level, we do not pretend to be able to evaluate the physics of the MJ79 model, as our measurements do not capture the microphysical scale of evaporation processes happening within the skin layer at the ocean-air interface. However, our measurements clearly reveal that - at a larger spatial scale - isotopic vapour data within the atmospheric boundary layer is not conform with the MJ79 model. Furthermore, we demonstrate that both simple isotope modeling (i.e. calculations based on the closure assumption) as well as more complex modelling approaches (isotopes within the atmosphere GCM ECHAM5-wiso) match better to the observations if the distinction between smooth and rough evaporative regimes is discarded. We have rephrased parts of the manuscript to clarify this difference between the MJ79 model approach and our own findings.

R3: I still miss a physical explanation for the wind speed treatment here. The use of Brutsaert is not a physical explanation, and the Reynolds number basically is a measure for laminar or turbulent flow. You say you cannot evaluate MJ79, but you do test their assumptions against data. Isn't this an evaluation?

Regarding the figure shown in our previous reply: It was never our intention to hide anything from the reader and we have now included the figure in the supplementary material, as suggested (new Supplementary Figure 4). We have added a discussion on this new figure in the text.

R3: I did not mean to imply that you hid anything on purpose, I am sorry if my formulation was misleading here. I think that this information is important and thus should be really given in the text. The formulation in the text is very short (l.166-169) and not well written and thus a bit hard to understand. Please rephrase.

I also have an additional point (I apologize for not having mentioned it in my first review):

You investigate the influence of sea ice on the stable isotope ratios and in the discussion of Greenland you nicely mention the possibility of “vapour blown from the Greenland ice sheet towards the research vessel”. However, this is the only case you include the influence of advection in your analysis. Antarctic sea ice is mainly found in the vicinity of the continent, which is strongly influenced by synoptic activity in the circumpolar trough, and close to the continent, can also be influenced by katabatic outflow. You never mention anything like that. You assume that the water vapour isotopology depends solely on local evaporation/sublimation conditions. Is that a valid assumption? I think it is not. It is most likely beyond the scope of the presented study to include an analysis of advection, however, these things should be mentioned in the discussion.

To evaluate the potential influence of an advection of air originating from the continent on our dataset used for the sea ice analyses, we applied an additional filter considering the wind direction and speed, as it was already done for the dataset representative of the open ocean. We removed from the sea ice dataset the periods where the observed wind was originating from a coastal region. Only a limited number of data points are removed by this additional filter (67 out of a total of 843 data points), as most of the areas of observations which are covered by sea ice are situated far from the coasts. As it can be seen in Figure A, such filtering does not affect the relationship between the sea-ice cover and the observed d-excess significantly. This information has been included in the manuscript. (b)

Figure A: Sensitivity of the deuterium excess in vapour with respect to different sea ice coverage conditions. (a) Same as Manuscript Figure 5, (b) with an additional filter removing any data points potentially influenced by wind originating from land-covered regions.

R3: I did not talk about advection about cold air from the continent, but about advection generally. Thus the removal of those 67 data points is not very helpful here and I also would not expect a different result here, given the total number of data points and the broad scattering. I can only repeat myself: You assume that the water vapor isotopology depends solely on local evaporation/sublimation conditions. Is that a valid assumption? I think it is not. It is most likely beyond the scope of the presented study to include an analysis of advection, however, these things should be mentioned in the discussion. You do mention that the closure equation is only valid on a global scale (l.161-162), I would just like to see a more explicit discussion of the consequences.

I also tried to figure out from Suppl. Fig. 3 whether increasing sea ice also generally meant increasing latitude (at least in Antarctica) (thus decreasing temperature, decreasing $\delta^{18}O$ etc.), but it was

impossible to get this information from the tiny figure that, particularly for Antarctica, has the sea ice information in a stretch hardly bigger than 1cm.

Around Antarctica, the sea ice cover is generally increasing when going southward. However, the latitudinal ship position and the sea ice cover surrounding the research vessel are not strongly correlated during our observational period (Fig. B). This might be explained by the fact that the ship usually tries avoiding the most compact ice regions to secure navigation. As pointed out by the reviewer, the sea ice cover map around Antarctica from Supplementary Figure 3.b was lacking of readability. Therefore, we improved this figure by zooming-in on smaller regions covered by sea ice.

Figure B: Sea ice cover (SIC) surrounding Polarstern, as a function of the latitude for the Southern Hemisphere only. The color indicates the observed air temperature.

R3: Nice figure, but I cannot find it in the paper or in the supplementary material. (see above)

I am also not sure if it makes sense to compare the entire open water data, which include the tropics and sub-tropics, to the sea ice data since those are completely different climatological regimes.

As stated in the Method section in the paragraph “Data filtering”, for the open-water related analyses

we have excluded any data points potentially influenced by sea ice coverage. Vice versa, the sea ice related analyses only include data points where sea ice surrounded the Polarstern. Thus, we believe that the interpretations of observations at open sea and in sea ice covered areas are treated in an adequate independent manner in this study.

R3: It is certainly very interesting to see the differences between open water and “sea ice covered” areas, even though I find the definition of “sea ice covered” a bit difficult. However, I was not talking about the difference between sea ice covered areas and open water, but about differences between open water in the tropics, the subtropics or the middle and high latitudes, which are very different in terms of climate. In the mid- and high latitudes the role of advection should be much larger than in the tropics and parts of the subtropics, this is the same point I mentioned earlier. I was not referring to cold air outbreaks from the continent only.

Given the simplifications you make by ignoring the influence of advection, it is a pretty strong statement that sublimation from snow on sea ice is the main source controlling the isotopic composition of the water vapor. (Sublimation needs more energy than evaporation, and from my understanding it involves isotopic fractionation, too.) Also, for most of the data points, sea ice concentration was smaller than 100% and even small openings in the ice are very important for heat and water/vapor exchange between ocean and atmosphere.

The modelling with the two assumptions, bare sea ice with the same isotopic composition as the ocean, and a snow surface on the seas ice, from which sublimation is possible, does not seem to be a convincing proof for the importance of the snow as source for the isotopic composition of the vapor. The empirical equation derived using the data does not give any physical explanation, but contains implicitly all processes that influence the isotopic composition of the vapor, both locally and advected. So, the isotopic composition of the snow is not independent of the isotopic composition of the vapor that is in the area after the snow was deposited. Thus, the result is not too surprising (you use the data to tune the model and then compare them to the data. Even though the data period is longer than the tuning period, I would not call this independent). I am not saying that your results are wrong, but just a bit too much simplified. As mentioned above, they should be formulated more cautiously and error possibilities due to atmospheric dynamic processes should be discussed.

Answers to Reviewer 3

“Water vapour isotopic composition over open ocean and sea ice in the Atlantic sector”

Note from the authors:

We have copied the 3rd review of Reviewer #3, which we received by email without any formatting, into this document. In accordance with the original formatting of Reviewer #3, his/her old response is printed in normal font, *our previous answer in italic*, **the latest comments by Reviewer #3 (marked by R3) in bold font**, and *our latest answer in blue font*.

Reviewer #3 (Remarks to the Author):

3rd review of Bonne et al., submitted to Nature Comm.

I marked my new comments with "R3", since my formatting got lost when I pasted my review into this field.

Generally I still think that this is a highly interesting paper, dealing with a unique data set, I only think that some aspects, especially simplifying assumptions, should be explained better and some results should be formulated more cautiously.

We thank Reviewer 3 again for his/her overall positive evaluation of our paper. We have included all recommendations in the revised manuscript and hope that we have dealt with all aspects in an adequate manner.

I will comment on the response to my latest review. My old response in normal font, the answer of the authors in italic, my present response in bold font.)

General remarks:

The authors did a thorough revision of the manuscript and addressed all points the reviewers had in their “Answers to the reviewers”. However, particularly in my case, they did not include the main points in the paper. If readers have the same comments as I had, they do not get the information the authors gave to the reviewer, which is not very satisfactory.

In case of publication, we intend to opt in for the Nature Communication scheme to publish all communication with the reviewers together with the article. Thus, the readers will have the same information as the reviewers.

R3: A publication should be self-contained and not many people have the time to read all the papers they want to read, let alone additionally all the correspondence with the reviewers of the papers. However, I leave it to the editor to decide whether this is the desired way to handle information. (From my point of view, even the supplementary information should not be necessary to basically understand the paper, but only for people with a deeper interest in the subject.)

We agree with Reviewer 3 that a publication should be self-contained and all the relevant information should be in the manuscript. Thus, we integrated the different topics of our discussion with Reviewer 3 in this revised version. In particular, we improved the discussion

regarding the absence of wind speed impact on our observed d-excess signal and discussed in more detail the potential influence of advection on our results and conclusions.

Specific comments:

Concerning the influence on wind speed on the deuterium excess:

I think generally you should make clear whether you are talking about testing the MJ79 findings or whether you are talking about general physics.

It would be good to give a physical explanation why the wind speed should influence d-excess according to MJ1979, and also possible explanations why it should not according to your findings. The interesting figure you showed to the reviewers should be included in the supplementary material and should be discussed in the text so that the reader can comprehend your statement. It is an important and valuable finding and you should not hide this from the reader.

From your investigation with the smooth and rough regime alone you can only state that not all findings of MJ79 are confirmed, but not that there is generally no dependence of d-excess on wind speed. (L144-146)

Merlivat and Jouzel (1979) derived a theoretical model to account for the deuterium-oxygen 18 relationship measured in meteoric waters. For the processes occurring during the evaporation of water, they have used results from a model by Brutsaert (1975). According to MJ79, this model was shown to be in good agreement with observed fractionation of the isotopic species of water occurring when water is evaporated and transported into the atmosphere under various laboratory (wind tunnel) conditions. Following the model of Brutsaert, in MJ79 a distinction between smooth and rough water- atmosphere interfaces is proposed. The domains of validity of these two interfaces are experimentally defined. A jump between both interfaces is proposed for surface roughness Reynolds numbers below or above 1.

As the content of this paper is based on experimental data taken within the atmospheric boundary layer at 29 m height above sea level, we do not pretend to be able to evaluate the physics of the MJ79 model, as our measurements do not capture the microphysical scale of evaporation processes happening within the skin layer at the ocean-air interface. However, our measurements clearly reveal that - at a larger spatial scale - isotopic vapour data within the atmospheric boundary layer is not conform with the MJ79 model. Furthermore, we demonstrate that both simple isotope modeling (i.e. calculations based on the closure assumption) as well as more complex modelling approaches (isotopes within the atmosphere GCM ECHAM5-wiso) match better to the observations if the distinction between smooth and rough evaporative regimes is discarded.

We have rephrased parts of the manuscript to clarify this difference between the MJ79 model approach and our own findings.

R3: I still miss a physical explanation for the wind speed treatment here. The use of Brutsaert is not a physical explanation, and the Reynolds number basically is a measure for laminar or turbulent flow. You say you cannot evaluate MJ79, but you do test their assumptions against data. Isn't this an evaluation?

The physical explanation for the wind speed dependency in the MJ79 model is that wind over a water surface will generate waves, which can change the strength of the evaporation flux. As evaporation is partly caused by molecular diffusion, which itself differs for H_2^{16}O and H_2^{18}O and $\text{H}^2\text{H}^{16}\text{O}$, the isotopic fractionation strength during evaporation might change with the occurrence of waves and thus with wind speed. In an experimental study, Merlivat (1978) artificially created waves in an air-water wind tunnel setup and found that the measured isotope vapour data were best represented by the evaporation model of Brutsaert (1975).

In the study of Merlivat (1978) the wind velocities could only vary from 0.7 to 6 m/s. Furthermore, the impact of waves has been evaluated only for a limited domain (non-breaking waves of 1Hz frequency with a maximum height of 12 cm). These conditions do not necessarily represent the diversity of surface oceanic conditions observed at sea and the surface roughness representation as a function of the local wind speed only might be too simplified. For example, a rough ocean surface with high waves might also be caused by swell, and does not have to be directly linked to high wind speeds occurring at the same time. Thus, oceanic evaporation might be more complex and diverse than in the experimental design of Merlivat (1978), which led to a distinction between smooth and rough surface regimes based on the model of Brutsaert (1975). Consequently, the wind speed dependency of the kinetic fractionation coefficients suggested in the MJ79 model might be an oversimplification of the real fractionation processes occurring in nature.

By saying that we cannot “*evaluate* the MJ79 model” we mean that we cannot state if this model is right or wrong as we have not performed (laboratory) experiments directly above the water surface as done in Merlivat 1978. The model might still be correct for the wind velocities and wave heights investigated in that study. However, by comparing our data with the MJ79 calculations we conclude that the model should not be *applied* in its original form for the calculation of isotopic changes in atmospheric vapour well above the ocean (e.g. as it is done in current isoGCMs). Based on our new data set, we rather suggest to modify the MJ79 model and use constant kinetic fractionation coefficients instead of wind-speed dependent values.

We have revised our manuscript regarding this topic and hope that the new wording is more understandable to the reader.

Regarding the figure shown in our previous reply: It was never our intention to hide anything from the reader and we have now included the figure in the supplementary material, as suggested (new Supplementary Figure 4). We have added a discussion on this new figure in the text.

R3: I did not mean to imply that you hid anything on purpose, I am sorry if my formulation was misleading here. I think that this information is important and thus should be really given in the text. The formulation in the text is very short (1.166-169) and not well written and thus a bit hard to understand. Please rephrase.

The discussion of the relationship between wind speed and deuterium excess for constant RH_{sea} and SST conditions has been rephrased and extended in the revised manuscript.

I also have an additional point (I apologize for not having mentioned it in my first review):

You investigate the influence of sea ice on the stable isotope ratios and in the discussion of Greenland you nicely mention the possibility of “vapour blown from the Greenland ice sheet towards the research vessel”. However, this is the only case you include the influence of

advection in your analysis. Antarctic sea ice is mainly found in the vicinity of the continent, which is strongly influenced by synoptic activity in the circumpolar trough, and close to the continent, can also be influenced by katabatic outflow. You never mention anything like that. You assume that the water vapour isotopology depends solely on local evaporation/sublimation conditions. Is that a valid assumption? I think it is not. It is most likely beyond the scope of the presented study to include an analysis of advection, however, these things should be mentioned in the discussion.

To evaluate the potential influence of an advection of air originating from the continent on our dataset used for the sea ice analyses, we applied an additional filter considering the wind direction and speed, as it was already done for the dataset representative of the open ocean. We removed from the sea ice dataset the periods where the observed wind was originating from a coastal region. Only a limited number of data points are removed by this additional filter (67 out of a total of 843 data points), as most of the areas of observations which are covered by sea ice are situated far from the coasts. As it can be seen in Figure A, such filtering does not affect the relationship between the sea-ice cover and the observed d-excess significantly. This information has been included in the manuscript.

Figure A: Sensitivity of the deuterium excess in vapour with respect to different sea ice coverage conditions. (a) Same as Manuscript Figure 5, (b) with an additional filter removing any data points potentially influenced by wind originating from land-covered regions.

R3: I did not talk about advection about cold air from the continent, but about advection generally. Thus the removal of those 67 data points is not very helpful here and I also would not expect a different result here, given the total number of data points and the broad scattering. I can only repeat myself: You assume that the water vapor isotopology depends solely on local evaporation/sublimation conditions. Is that a valid assumption? I think it is not. It is most likely beyond the scope of the presented study to include an analysis of advection, however, these things should be mentioned in the discussion. You do mention that the closure equation is only valid on a global scale (1.161-162), I would just like to see a more explicit discussion of the consequences.

To compare our observations to the theoretical calculations based on MJ79, we only consider the locally measured values of relative humidity, temperature and oceanic isotopic composition, without considering any mixing with advected air masses. This is indeed a strong assumption. Looking at the first-order isotope values $\delta^{18}\text{O}$ and $\delta^2\text{H}$ indicates indeed that these calculations based on MJ79 do not allow to reproduce the observed isotopic signal in most places (Supplementary Figure 5). Despite this mismatch on $\delta^{18}\text{O}$ and $\delta^2\text{H}$, the MJ79-

based isotope values reproduce the locally observed d-excess signal and allow us to conduct sensitivity tests on this parameter (Supplementary Figure 6).

For the comparison of our observations with simulation results of the isoGCM ECHAM5-wiso model, the influence of advection is not neglected. Potential changes of the isotopic composition of water vapor by advection are explicitly simulated by the isoGCM.

As suggested, we have rephrased and expanded the discussion of this important topic in the manuscript.

I also tried to figure out from Suppl. Fig. 3 whether increasing sea ice also generally meant increasing latitude (at least in Antarctica) (thus decreasing temperature, decreasing $\delta^{18}\text{O}$ etc.), but it was impossible to get this information from the tiny figure that, particularly for Antarctica, has the sea ice information in a stretch hardly bigger than 1cm.

Around Antarctica, the sea ice cover is generally increasing when going southward. However, the latitudinal ship position and the sea ice cover surrounding the research vessel are not strongly correlated during our observational period (Fig. B). This might be explained by the fact that the ship usually tries avoiding the most compact ice regions to secure navigation. As pointed out by the reviewer, the sea ice cover map around Antarctica from Supplementary Figure 3.b was lacking of readability. Therefore, we improved this figure by zooming-in on smaller regions covered by sea ice.

Figure B: Sea ice cover (SIC) surrounding Polarstern, as a function of the latitude for the Southern Hemisphere only. The color indicates the observed air temperature.

R3: Nice figure, but I cannot find it in the paper or in the supplementary material. (see above)

We have modified the Supplementary Figure 9 to present the same information as Figure B (sea ice cover relationship with and air temperature and latitude) under the same consistent manner as the other parameters ($\delta^{18}\text{O}$, d-excess, RH_{sea} and SST). This additional information is also discussed in the manuscript.

I am also not sure if it makes sense to compare the entire open water data, which include the tropics and sub-tropics, to the sea ice data since those are completely different climatological regimes.

As stated in the Method section in the paragraph “Data filtering”, for the open-water related analyses we have excluded any data points potentially influenced by sea ice coverage. Vice versa, the sea ice related analyses only include data points where sea ice surrounded the

Polarstern. Thus, we believe that the interpretations of observations at open sea and in sea ice covered areas are treated in an adequate independent manner in this study.

R3: It is certainly very interesting to see the differences between open water and “sea ice covered” areas, even though I find the definition of “sea ice covered” a bit difficult. However, I was not talking about the difference between sea ice covered areas and open water, but about differences between open water in the tropics, the subtropics or the middle and high latitudes, which are very different in terms of climate. In the mid- and high latitudes the role of advection should be much larger than in the tropics and parts of the subtropics, this is the same point I mentioned earlier. I was not referring to cold air outbreaks from the continent only.

We thank Reviewer 3 for this clarification of his/her previous comment. We agree with the reviewer on the potentially varying importance of advection in different regions. The comparison of our observations with the theoretical values based on MJ79 might indeed reveal the large role of advection in mid- and high-latitude regions as compared to (sub)tropical regions. As shown in Supplementary Figure 5, the difference between observed $\delta^{18}\text{O}$ and $\delta^2\text{H}$ values as compared to the MJ79-based calculations is low in the tropics, and much larger at mid- to high-latitudes. This could be explained by different relative contributions of locally evaporated vapour versus advected moisture in (sub-)tropical versus mid- to high-latitude regions. Despite this varying misfit of $\delta^{18}\text{O}$ and $\delta^2\text{H}$, the MJ79-based calculations show a comparable good agreement with the observed d-excess values in all different regions. Based on this finding we decided to combine all our measurements from open water areas from different latitudinal regions for the analysis of the meteorological parameters controlling the d-excess in vapor.

Given the simplifications you make by ignoring the influence of advection, it is a pretty strong statement that sublimation from snow on sea ice is the main source controlling the isotopic composition of the water vapor. (Sublimation needs more energy than evaporation, and from my understanding it involves isotopic fractionation, too.) Also, for most of the data points, sea ice concentration was smaller than 100% and even small openings in the ice are very important for heat and water/vapor exchange between ocean and atmosphere.

It is certainly true that evaporation from open water will happen more easily than sublimation from snow on sea ice. In areas, where the sea ice fraction is less than 100%, evaporation must not be neglected. However, evaporation from seawater (with an isotopic value close to 0‰) cannot explain the measured very low vapour isotope values. Thus, we think that sublimation of snow on sea ice is a key additional process controlling the isotopic composition of the water vapor in the polar regions. Without considering this process, our instrumental data cannot be explained. We have rephrased the manuscript, accordingly.

We are also aware of the possible fractionation occurring during sublimation. As the actual reason and strength of this fractionation process is not well understood, yet (e.g. Steen-Larsen et al., 2014), we did not consider it in our model approach. We now discuss this topic in the revised manuscript.

The modelling with the two assumptions, bare sea ice with the same isotopic composition as the ocean, and a snow surface on the sea ice, from which sublimation is possible, does not seem to be a convincing proof for the importance of the snow as source for the isotopic composition of the vapor. The empirical equation derived using the data does not give any physical explanation, but contains implicitly all processes that influence the

isotopic composition of the vapor, both locally and advected. So, the isotopic composition of the snow is not independent of the isotopic composition of the vapor that is in the area after the snow was deposited. Thus, the result is not too surprising (you use the data to tune the model and then compare them to the data. Even though the data period is longer than the tuning period, I would not call this independent). I am not saying that your results are wrong, but just a bit too much simplified. As mentioned above, they should be formulated more cautiously and error possibilities due to atmospheric dynamic processes should be discussed.

It is correct that our tuning of the model is not fully independent of the data, and that the suggested parametrization is simplified. We rate it as a first-order approach to include snow on sea ice for future isotope modelling studies with GCMs, only. Further studies are certainly necessary to improve this parametrization and we have reformulated the uncertainty and error possibilities of our approach, as recommended by the reviewer.

References

1. Merlivat, L. The dependence of bulk evaporation coefficients on air-water interfacial conditions as determined by the isotopic method. *J. Geophys. Res.* **83**, 2977–2980 (1978).
2. Brutsaert, W. A theory for local evaporation (or heat transfer) from rough and smooth surfaces at ground level. *Water Resour. Res.* **11**, 543–550 (1975).
3. Steen-Larsen, H. C. *et al.* What controls the isotopic composition of Greenland surface snow? *Clim. Past* **10**, 377–392 (2014).

Reviewer 3:

I would say the paper is ready for publication now, I only have a small comment: line 192-194 in the version with the marked changes (page 8, end of first paragraph) do not make sense. The authors should rephrase this or keep their earlier version here (explanation of the overestimation of isotope ratios by the model due to the closure assumption, which is most likely not valid here.)